# Electrophysiological mechanisms of human memory consolidation

Hui Zhang[1], Juergen Fell[2] & Nikolai Axmacher[1]

Consolidation stabilizes memory traces after initial encoding. Rodent studies suggest that memory consolidation depends on replay of stimulus-specific activity patterns during fast hippocampal "ripple" oscillations. Here, we measured replay in intracranial electro-encephalography recordings in human epilepsy patients, and related replay to ripples. Stimulus-specific activity was identified using representational similarity analysis and then tracked during waking rest and sleep after encoding. Stimulus-specific gamma (30–90 Hz) activity during early (100–500 ms) and late (500–1200 ms) encoding is spontaneously reactivated during waking state and sleep, independent of later memory. Ripples during nREM sleep, but not during waking state, trigger replay of activity from the late time window specifically for remembered items. Ripple-triggered replay of activity from the early time window during nREM sleep is enhanced for forgotten items. These results provide the first electrophysiological evidence for replay related to memory consolidation in humans, and point to a prominent role of nREM ripple-triggered replay in consolidation processes.

[1] Department of Neuropsychology, Institute of Cognitive Neuroscience, Faculty of Psychology, Ruhr University Bochum, Bochum 44801, Germany.
[2] Department of Epileptology, University of Bonn, Bonn 53105, Germany. Correspondence and requests for materials should be addressed to H.Z. (email: Hui.Zhang-c5u@ruhr-uni-bochum.de) or to N.A. (email: nikolai.axmacher@ruhr-uni-bochum.de)

Only a minority of the information we encounter each day will be remembered on the long term, and this depends crucially on memory consolidation. Rodent studies suggest that hippocampal "ripple" oscillations (around 200 Hz)[1,2] and replay of stimulus-specific neural activity[3–5] serve as neurophysiological mechanisms underlying memory consolidation. In rodents, ripples and replay are temporally coupled[6]. Further studies suggest that ripple-triggered replay plays a causal role for memory consolidation, because disrupting ripples impairs memory performance after sleep[2,7].

In humans, there is abundant evidence that brain regions which are involved in cognitive tasks before a period of sleep are more active during sleep, and that the activation levels of these regions are related to subsequent memory performance[8,9]. Moreover, when participants encoded visual stimuli together with odors or auditory cues and these cues were presented to them again during subsequent sleep, memory after the sleep was enhanced[10–13]. Areas of the medial temporal lobe including the hippocampus[10] and parahippocampal gyrus[12] showed increased blood oxygen level dependent (BOLD) responses in functional magnetic resonance imaging (fMRI) when the cues were presented during sleep, consistent with reactivation of previously acquired memory traces. More recently, several studies used fMRI to measure replay of stimulus-specific activity patterns[14,15]. In these studies, stimulus-specific representations were identified during encoding, using either multivariate pattern classification[16] or representational similarity analysis (RSA)[17]. Replay was then assessed via the spontaneous reoccurrence of stimulus-specific activity across subsequent resting periods. Replay was observed during both waking state and sleep, in apparent conflict with behavioral data showing that sleep—in particular, non-rapid eye movement (nREM) sleep—benefits memory consolidation[18–22]. These studies provided first evidence for replay in humans and showed that the amount of replay was related to subsequent memory performance[14,15,23].

In line with evidence from rodent studies, ripples are functionally relevant for memory consolidation in humans as well[24]. However, due to the low temporal resolution of fMRI, these previous studies were not able to evaluate the relationship between replay and ripples. In humans, ripples can only be recorded in presurgical epilepsy patients implanted with intracranial EEG (iEEG) electrodes[24–27]. Several studies have shown that ripples in humans have a lower frequency than in rodents (around 100 Hz), possibly because of the involvement of larger networks[28]. However, direct electrophysiological evidence for replay related to memory consolidation in humans is still missing.

Here, we first identified stimulus-specific neural representations in humans by applying a previously established RSA-based metric[29–31] to iEEG data from 12 epilepsy patients. We then tracked replay of stimulus-specific activity during a post-encoding afternoon nap (containing awake resting state and sleep periods), and related replay to hippocampal ripple events. Our results show that stimulus-specific information is spontaneously replayed during both waking state and nREM sleep, but these general replay levels do not predict later memory. By contrast, ripple-triggered replay of activity from late encoding stages (500–1200 ms after stimulus onset) occurs specifically for items that are later remembered, and only during nREM sleep. These findings suggest a mechanistic explanation for the beneficial role of sleep for memory consolidation.

## Results

### Stimulus-specific representations during encoding. To assess stimulus-specific engram patterns, we first extracted for each channel and each item the iEEG activity within the gamma and epsilon band (30–90 Hz and 90–150 Hz, respectively; see Supplementary Note 2; Supplementary Fig. 2a, b; Methods). We used consecutive time windows of 200 ms (autocorrelation of the gamma activity was significant up to 250 ms; Supplementary Note 4; Supplementary Fig. 2e), overlapping by 100 ms (Fig. 1c, results refer to the center time points of the windows; we also tried consecutive time window of 100 ms, overlapping by 50 ms and it generated highly similar results; Supplementary Fig. 2c; Supplementary Note 3). During each time window, we obtained one activity pattern across electrodes per item. These patterns were correlated between each encoding time window and each retrieval time window of each item (Fig. 1b). Stimulus-specific representations were identified by comparing correlations between encoding and retrieval of the same item ($RSA_{same}$) to correlations between encoding of one and retrieval of a different item ($RSA_{differ}$). We controlled for multiple comparisons using surrogate-based cluster statistics[30].

Consistent with previous results[29,30], we observed significantly higher $RSA_{same}$ than $RSA_{differ}$ values in the gamma frequency range (30–90 Hz). Two separate temporal clusters were identified (Fig. 2a) with regard to encoding, an early cluster (100–500 ms, blue frame) and a late cluster (500–1200 ms, black frame; both $p_{corr} < 0.001$). $RSA_{same}$ values were higher for remembered than forgotten items in the late ($t(11) = 3.75$, $p = 0.003$) but not the early cluster ($t(11) = 0.45$, $p = 0.66$; Fig. 2b), indicating a relevance for subsequent memory only for the late cluster. In order to investigate in greater detail which encoding time windows are relevant for later memory, we compared the difference of the similarity levels between remembered and forgotten items within clusters showing stimulus-specific representations and found that predominantly reinstatement of activity during late time window (800–1100 ms; capturing activity in the time range from 700 to 1200 ms) was relevant for subsequent memory ($p < 0.001$ after multiple comparison correction; surrogate data were generated by shuffling condition labels of trial averages in individual participants). $RSA_{same}$ values did not differ between the nap day and the day without nap for either remembered or forgotten items (all $t(11) < 0.68$, all $p > 0.51$). Two clusters in the epsilon band (90–150 Hz) showed higher $RSA_{same}$ than $RSA_{differ}$ values as well (Fig. 2c: 300–700 ms and 500–900 ms), however these effects were unrelated to memory (all $t(11) < 0.84$, all $p > 0.52$; Fig. 2d). We further investigated the brain regions underlying stimulus-specific representations in both time clusters. We found that 15% of all electrodes contributed significantly to stimulus-specific representations of either the early or the late encoding cluster (Supplementary Fig. 2d) which replicated our previous findings[30] supporting the theoretical notion of sparse coding. We further investigated the distribution of positively contributing electrodes and found that more electrodes showed significant positive contributions to the early encoding cluster than the late encoding cluster ($\chi^2(1) = 5.69$, $p = 0.017$) in the lateral temporal lobe (including fusiform gyrus, inferior temporal gyrus, and middle temporal gyrus). We found a trend for an opposite pattern for electrodes within the medial temporal lobe (MTL, including the hippocampus and the parahippocampal gyrus) where more electrodes contributed significantly to the late cluster than to the early cluster ($\chi^2(1) = 2.96$, $p = 0.085$). Due to the overall small number of electrodes in the occipital (2 electrodes) and parietal (2 electrodes) cortex, we did not perform statistical analyses in these regions.

### Spontaneous replay during rest. We assessed the spontaneous re-occurrence of stimulus-specific activity patterns of remote items (presented on the nap day before rest) during the rest period by calculating the similarity of these patterns during

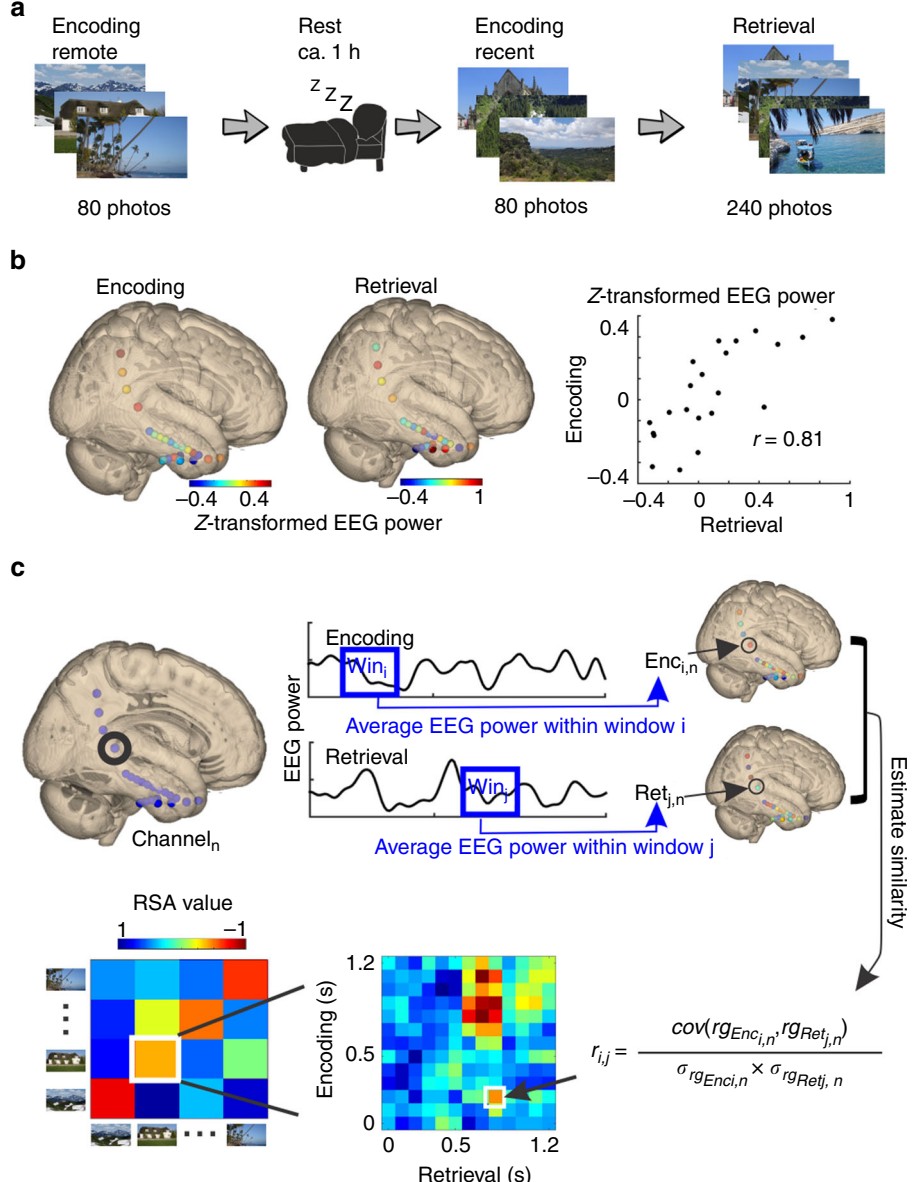

**Fig. 1** Analysis of stimulus-specific representations. **a** Experimental design (day with nap). **b** Example of distributed activity in the gamma frequency range (30–90 Hz) during encoding and retrieval of the same item. The right panel shows the same example data as a scatter plot. Each sphere corresponds to one electrode contact. **c** Analysis of stimulus-specific representations: schematic overview. EEG power time series were extracted during encoding and retrieval of each item for each channel and each frequency band. EEG data were segmented into 200 ms windows with 100 ms overlap, and averaged within each window and each channel for both encoding and retrieval. This resulted in a 13 (window) × n (channel) × 2 (encoding, retrieval) matrix for each item and each patient. We then performed Spearman's correlations across channels separately for each window during encoding and each window during retrieval of the same item. This resulted in one 13 (encoding window) × 13 (retrieval window) correlation matrix for each item and each patient. Fisher-Z transformed correlation matrices were averaged across items within each patient, resulting in one 13 (encoding window) × 13 (retrieval window) correlation matrix (RSA$_{same}$) for each patient. A similar analysis was conducted between encoding of one item and retrieval of a different item. This resulted in another 13 (encoding window) × 13 (retrieval window) correlation matrix (RSA$_{differ}$) for each patient. We then performed paired t tests between RSA$_{same}$ and RSA$_{differ}$ across patients. We used a cluster-based surrogate analysis for correction of multiple comparisons. Surrogate data were generated by randomly shuffling item labels and performing the same RSA procedure as described above. We extracted clusters from these surrogate matrices and selected for each surrogate the cluster with the largest summary t value. This surrogate procedure was repeated 10,000 times and produced 10,000 surrogate clusters. Then, we ranked each empirical cluster within the distribution of all surrogate clusters. rg indicates the ranking value. For details, see Methods

encoding with brain activity during rest (Methods and Fig. 3a). These reactivations were compared to correlations between rest activity and recent items (shown on the nap day after rest) which served as a baseline (Supplementary Note 1)[14]. We analyzed both spontaneous replay (across the entire rest period; Fig. 3a) and replay coincident with hippocampal ripples. Due to the small number of patients reaching REM sleep (5 out of 12) and the

overall short duration of REM sleep in our data (Supplementary Fig. 3a), we focused on waking state and nREM sleep (including sleep stages N1, N2, and N3). The duration did not differ between waking state and nREM sleep ($t(11) = 0.82$; $p = 0.43$).

To analyze spontaneous replay, we first calculated the level of replay of gamma band activity during each 200 ms time window (overlapping by 100 ms) during encoding. We averaged replay

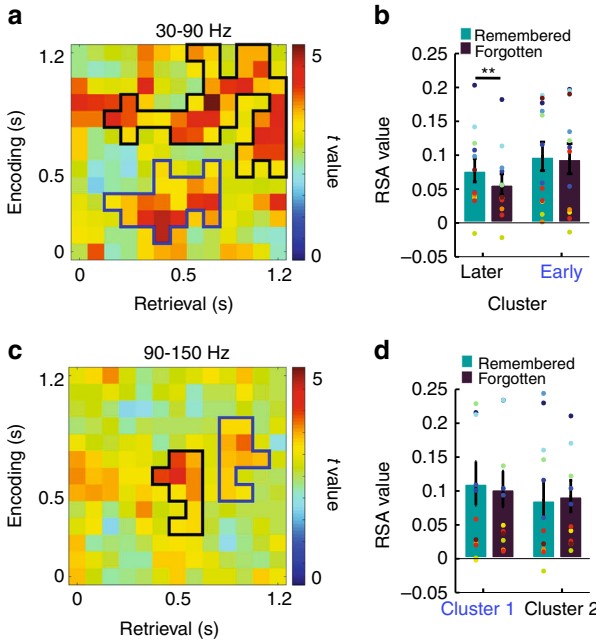

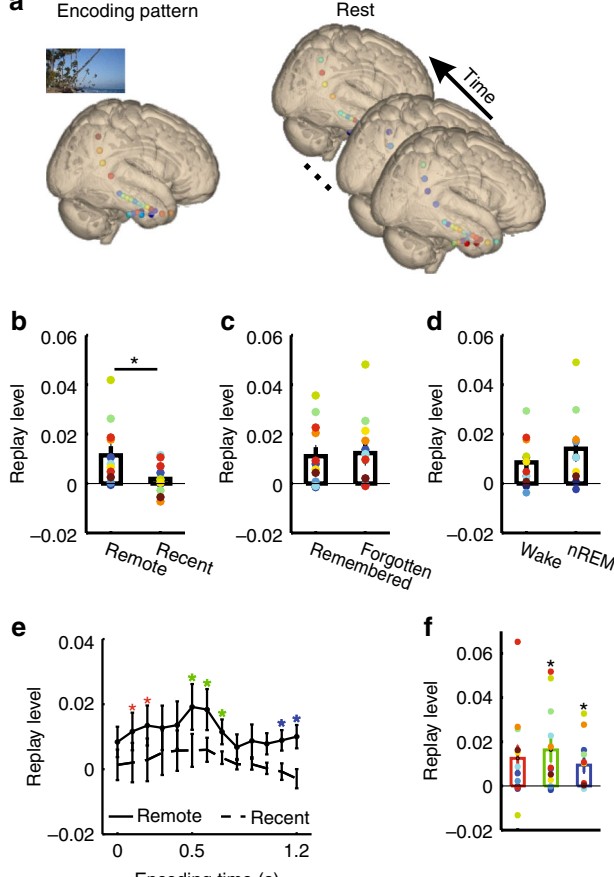

**Fig. 2** Identification of stimulus-specific activity patterns. **a** Two clusters showed stimulus-specific representations (i.e., higher correlations between encoding of one and retrieval of the same item as compared to encoding of one and retrieval of a different item) in the gamma range: an early cluster (100–500 ms, blue frame) and a late cluster (500–1200 ms, black frame). **b** Functional relevance of clusters showing stimulus-specific representations in panel a for memory. Higher similarity between encoding and retrieval of subsequently remembered than forgotten items in the late cluster, but not in the early cluster. Each colorful dot indicates one participant. Same colors indicate data from the same participant. **c** Two clusters showed stimulus-specific representations in the epsilon range. **d** The similarity level within clusters in **c** did not differ between remembered and forgotten items. Each colorful dot indicates one participant. Same colors indicate data from the same participant. Error bars, standard error of the mean; **p < 0.01 (paired t test)

**Fig. 3** Spontaneous replay during rest. **a** Schematic illustration of analysis procedure. Reactivation is quantified by correlating activity during each encoding event and each time period during the resting state. Each circle in the brain indicates one channel. The hotter the color of the circle, the higher the EEG power. **b** Higher replay of stimulus-specific activity from late encoding cluster (500–1200 ms) of remote vs. recent items. * indicates p < 0.05 (paired t test). Reactivation does not differ between remembered and forgotten remote items **c** or between waking state and nREM sleep **d**. **e** Encoding-time resolved replay during rest. Reactivation differs between remote and recent items in three time periods marked with *s in different colors after cluster-based correction for multiple comparisons (surrogate data was generated by switching remote and recent labels). **f** Averaged replay levels of all clusters depicted in **e** as compared to zero. Each colorful dot indicates one participant. Same colors indicate data from the same participant. Error bars, standard error of the mean; * indicates significant difference from zero (one sample t-test, p < 0.05)

levels first across the eight time windows of the late (500–1200 ms) encoding cluster for each item, and then across items. This was done separately for remote and recent items. Replay levels were significantly larger than zero for remote ($t(11) = 3.07$, $p = 0.011$) but not recent items ($t(11) = 1.11$, $p = 0.29$; Fig. 3b). Moreover, replay levels of remote items were significantly higher than those of recent items ($t(11) = 2.32$, $p = 0.041$). We next separately analyzed replay during the different states of vigilance. Replay levels of remote items were significantly above zero during both waking state ($t(11) = 3.11$, $p = 0.010$) and nREM sleep ($t(9) = 2.88$, $p = 0.018$). This was not the case for recent items (both $t(11) < 1.31$, both $p > 0.34$). Replay levels were higher for remote than recent items during waking state ($t(11) = 2.33$, $p = 0.040$) and marginally during nREM sleep ($t(9) = 1.85$, $p = 0.098$). Replay levels of remote items did not differ between waking state and nREM sleep ($t(9) = 1.46$, $p = 0.18$; Fig. 3d). For gamma band activity from the early encoding cluster only a marginally significant replay effect was found (100–500 ms; Supplementary Fig. 4a; Supplementary Note 5).

We also tested for replay of the entire sequence (i.e., the temporal evolution) of gamma band activity within the two encoding clusters showing stimulus-specific representations. Results were very similar to replay of activity from the individual encoding time windows during these clusters described above (Supplementary Note 6; Supplementary Fig. 5). We did not find

any temporally compressed or expanded replay effects (Supplementary Note 9).

Next, we analyzed replay of gamma band activity from all encoding time windows (i.e., extending beyond the two clusters showing stimulus-specific representations). Replay was analyzed for all 200 ms time windows, overlapping by 100 ms, covering the entire 1200 ms period from the onset to the offset of stimulus presentation. Several encoding time windows showed higher replay levels for remote vs. recent items (100–200 ms, 500–700 ms, and 1100–1200 ms; Fig. 3e; all $p < 0.035$, cluster-corrected for multiple comparisons; surrogate data was generated by switching remote and recent labels). Reactivations of remote items were significantly larger than zero during the latter two time periods (500–700 ms: $t(11) = 2.98$, $p = 0.013$; 1100–1200 ms: $t(11) = 2.84$, $p = 0.016$; Fig. 3f) and in trend during the first time period

(100–200 ms: $t(11) = 2.14$, $p = 0.056$). We also separately compared replay levels of remote and recent items during different stages of vigilance (waking state and nREM sleep) for all encoding time windows. During waking state and nREM sleep, replay levels of remote items were higher than those of recent items for late windows (waking stage, 1000–1200 ms; $p = 0.022$; nREM sleep, 500–700 ms; $p = 0.021$; $p$ values were corrected for multiple comparisons; surrogate data was generated by switching remote and recent labels; Supplementary Fig. 4d, e, left panels). Replay levels of remote items were consistently larger than zero in both clusters (waking state: $t(11) = 2.59$, $p = 0.025$; nREM sleep: $t(9) = 3.42$, $p = 0.0076$; Supplementary Fig. 4d, e, right panels). These findings are thus consistent with the results of the analysis focusing on the late encoding cluster and show that stimulus-specific activity occurring relatively late after stimulus presentation is spontaneously reactivated during subsequent rest periods.

Is spontaneous replay of gamma band activity related to subsequent memory? We did not find different replay levels when comparing remembered and forgotten remote items, for either the late (500–1200 ms; Fig. 3c) or the early (100–500 ms; Supplementary Fig. 4a) encoding cluster (both $t(11) < 0.63$, both $p > 0.55$).

In the epsilon band, we did not find any evidence for replay, either for activity during the two clusters showing stimulus-specific epsilon band activity during encoding (300–700 ms and 500–900 ms; Supplementary Note 7; Supplementary Fig. 4b–c) or for activity sequences during each cluster (Supplementary Note 8; Supplementary Fig. 5d–e). Replay levels did not differ between remembered and forgotten items for either encoding cluster (both $t(11) < 0.20$, both $p > 0.85$; Supplementary Fig. 4b–c; Supplementary Fig. 5d–e).

Together, these results provide an electrophysiological basis for previous fMRI findings showing replay of stimulus-specific activity[14,15]. More specifically, they show that spontaneous replay is predominantly related to gamma band activity occurring relatively late (>500 ms) after stimulus presentation. In line with previous fMRI studies on replay[14,15], similar replay levels were observed during waking state and nREM sleep.

Control analyses ruled out that replay was related to the overall power of 30–90 Hz activity during encoding of an item (Supplementary Fig. 6a–b; Supplementary Note 10) or during the rest period (Supplementary Fig. 6c; Supplementary Note 11). Numerically, replay levels were even higher when resting gamma power was low, as shown by negative correlation coefficients between replay levels and resting gamma power ($r = -0.14 \pm 0.30$; mean ± SD), but correlations were not consistently different from zero ($t(11) = -1.61$, $p = 0.14$). Remote items showed similar replay levels across the entire rest period, excluding a possible bias caused by temporal correlations between brain activity during encoding and rest (Supplementary Fig. 6d and Supplementary Note 12).

**Ripple-triggered replay.** Our data reported so far demonstrate that items presented before the rest period ("remote") are replayed more often than control items presented afterwards ("recent"). We next scrutinized replay of remote items in temporal relationship to hippocampal ripple oscillations, focusing on stimulus-specific activity in the gamma frequency range that showed spontaneous replay. We extracted hippocampal ripples based on previously established criteria (Methods; Fig. 4a, b and Supplementary Fig. 7). The incidence of ripples (mean ± SD: 3.04 ± 0.70 events per minute) was comparable to previous reports[24,32]. The number of ripple events did not differ between waking state and nREM sleep ($t(11) = 1.37$; $p = 0.20$). Figure 4d displays ripple-triggered replay levels of gamma band activity

during encoding, which we compared to replay during various "peri-ripple periods", i.e., time intervals before and after ripples (Methods; Fig. 4c). We found that during nREM sleep but not during waking state, replay was enhanced during ripple events as compared to peri-ripple periods (Fig. 4d and Supplementary Fig. 8c). In addition, the ripple-locked replay level of remote items was higher than the ripple-locked replay level of recent items for both the early and late cluster during nREM sleep (both $t(9) > 2.19$, both $p < 0.016$). This was not the case during waking state (both $t(11) < 0.81$, both $p > 0.43$).

**Ripple-triggered replay of remote items during nREM sleep.** We first evaluated nREM sleep ripple-locked replay of activity from the early (100–500 ms) and late encoding cluster (500–1200 ms) for both remembered and forgotten items. We performed a $2 \times 2$ ANOVA with "cluster" and "memory" as repeated measures and found a significant interaction ($F(1,9) = 11.12$, $p = 0.0087$; Fig. 4e). Further analyses showed that replay of activity from the late cluster was higher than replay of activity from the early cluster for remembered items ($t(9) = 2.92$, $p = 0.017$), while the pattern was reversed for forgotten items ($t(9) = 2.04$, $p = 0.071$).

For subsequently remembered remote items, encoding activity from the late but not the early cluster was more replayed during ripple events than during the overall (averaged) peri-ripple periods (late cluster: $t(9) = 2.61$, $p = 0.028$; early cluster: $t(9) = 1.31$, $p = 0.22$). An ANOVA comparing replay of activity from the late cluster during ripple events and all ten peri-ripple periods yielded the same result ($F(10,90) = 3.94$, $p < 0.001$; Supplementary Fig. 8a). Post hoc $t$ tests showed that late-cluster replay was higher during ripple events as compared to all individual peri-ripple time windows, except the two windows immediately following the ripple (Supplementary Fig. 8a). Late-cluster replay levels for remembered items during ripple events were significantly greater than zero ($t(9) = 2.75$, $p = 0.022$). Moreover, the replay level of the late cluster was higher than both randomly selected epochs of spontaneous replay (Supplementary Note 13) and replay locked to surrogate ripples (Supplementary Note 14).

For subsequently forgotten remote items, we observed a strikingly different pattern of results. Replay of activity from the late cluster did not differ between ripple events and peri-ripple periods ($t(9) = 1.40$, $p = 0.20$). Interestingly, though—and in contrast to the results for later remembered items—replay of activity from the early cluster was higher during ripple events than during peri-ripple periods ($t(9) = 2.73$, $p = 0.023$). An ANOVA comparing replay of activity from the early cluster during ripple events and all ten peri-ripple periods showed a similar effect ($F(10,90) = 4.99$, $p < 0.001$; Supplementary Fig. 8a). Post hoc $t$ tests indicated that the replay level of the early activity from forgotten items was higher during ripple events compared to all individual peri-ripple time windows, except the two windows immediately following the ripple. Early cluster replay levels for forgotten items during ripple events were significantly greater than zero ($t(9) = 3.29$, $p = 0.009$). The replay level of the early cluster was higher than both randomly selected epochs of spontaneous replay (Supplementary Note 13) and replay locked to surrogate ripples (Supplementary Note 14).

Analogous to the analysis of spontaneous replay, we also compared replay during ripple events and peri-ripple periods separately for all individual encoding time windows for both remembered and forgotten items. Encoding-related activity between 700 and 800 ms was significantly more replayed during ripple events vs. peri-ripple periods for subsequently remembered remote items ($p = 0.041$, cluster-corrected for multiple

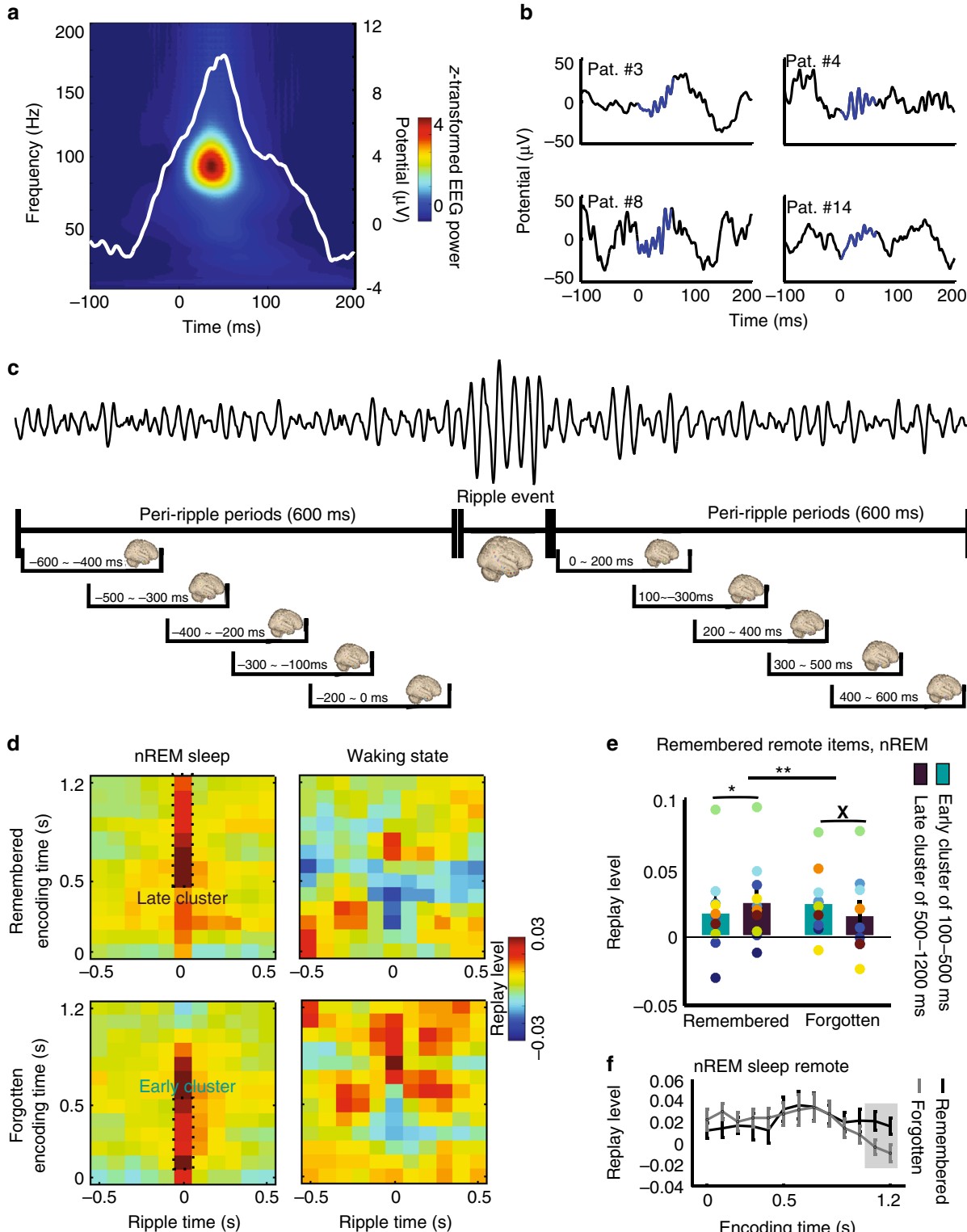

comparisons; Supplementary Fig. 8b) during nREM sleep. We did not find any other effects either for forgotten items during nREM sleep or for remembered or forgotten items during waking state.

Finally, we also directly compared replay levels of remembered and forgotten remote items within all individual encoding time windows. The replay level of activity from a late encoding time period (1100–1200 ms) was higher for remembered than forgotten items ($p = 0.014$, cluster-corrected for multiple comparisons; Fig. 4f), supporting the relevance of ripple-triggered

replay of late encoding activity during nREM sleep for memory consolidation.

**Control analyses for ripple-triggered replay.** During waking state, results were clearly different (Fig. 4d, right column): Replay of activity from the early and the late cluster did not differ between ripple events and peri-ripple periods, either for remembered or forgotten remote items (Supplementary Note 15).

**Fig. 4** Ripple-triggered replay. **a** Grand average of ripples and ripple-triggered raw EEG traces in the hippocampus. Time 0 indicates the onset of ripple events. **b** Examples of individual ripple events in four patients (highpass filtered at 5 Hz for display purposes). The blue line marks the duration of ripple events. **c** Analysis of replay during ripple events and peri-ripple periods: schematic overview. Top: Example of intracranial EEG activity from the hippocampus (highpass filtered at 40 Hz). Below: time windows used to extract activity during peri-ripple periods and ripple events. **d** Time resolved replay of gamma activity locked to ripples during nREM sleep and waking state for later remembered and forgotten remote items. Ripple time 0 corresponds to ripple events and ripple time before and after 0 correspond to peri-ripple period. The early and the late cluster correspond to the stimulus-specific clusters in Fig. 2a. **e** Ripple-triggered replay of early and late encoding activity differentially affects memory: Interaction between replay levels of remembered and forgotten items and gamma band activity from the early (100–500 ms) and the late (500–1200 ms) encoding clusters. **f** Direct comparisons of replay levels during ripple events between later remembered and later forgotten remote items. Each colorful dot indicates one participant. Same colors indicate data from the same participant. Error bars, standard error of the mean; *$p < 0.05$ (paired $t$ test); **$p < 0.01$ ($2 \times 2$ repeated measures ANOVA); X, $p = 0.071$ (paired $t$ test); time windows showing higher replay levels for remembered vs. forgotten items during ripple events are indicated by gray background

Replay levels of recent items did not differ between ripple events and peri-ripple periods during either nREM sleep or waking state (Supplementary Fig. 8c and Supplementary Note 16).

Control analyses ruled out that the observed higher replay level of stimulus-specific gamma band activity during ripple events compared with peri-ripple periods during nREM sleep was driven by power differences in either the ripple (80–100 Hz) or the gamma (30–90 Hz) frequency bands (Supplementary Notes 17–18). After matching the ripple power between nREM sleep and waking state, the ripple-triggered replay effect was still restricted to nREM sleep (Supplementary Note 17). We also did not find any relation between gamma power and ripple-locked replay levels (Supplementary Note 18), which is consistent with our findings on spontaneous replay.

## Discussion

Our results provide direct electrophysiological evidence for replay of stimulus-specific neural representations in humans. They show that replay is related to hippocampal ripples, and that ripple-triggered replay occurs specifically during nREM sleep. These data are conceptually consistent with previous findings of replay during ripples in rodents[4,33]. Unlike the single-cell results in rodents, however, stimulus-specific representations were analyzed on a network level. The relationship between stimulus-specific representations at the level of single units and in large-scale brain networks remains to be addressed in future studies in both animals and humans[34].

Notably, in the rodent literature, researchers typically record activities from a control sleep period before the task[35]. However, in humans a bias between the first and the second sleep period may occur because of the typical physiological adaptations to sleep laboratory conditions. In addition, we did not want to overstrain the patients by fixing polysomnography electrodes two times. On the other hand, two sleep periods back in back with a learning session in between was logistically not possible because we used an experimental design with an afternoon nap, and patients would not have been able to sleep both before and after the experiment. Instead of a control nap before the experiment to which replay during the experimental nap would be compared, we presented stimuli both before and after the nap and compared replay of remote and recent items. The presentation of stimuli both before and after the sleep period optimally balances temporal distances and thereby prevents any possible problems related to temporal autocorrelations. Further studies are required to assess the difference between those two different experimental settings.

Consistent with previous results[29,30], stimulus-specific representations were found predominantly in the gamma frequency range. Previous studies combining electrophysiological recordings with fMRI in monkey visual cortex reported a positive relationship between gamma power and BOLD responses[36]. Thus, it may

appear surprising that we found slightly negative correlations between replay scores and gamma power during rest. Even though this relationship was not significant (and is therefore difficult to interpret), it is reminiscent of a previous finding from our group that encoding-retrieval similarity during viewing of movies is negatively related to gamma power during retrieval in a significant subset of electrodes[30]. This was particularly the case for "informative" electrodes, i.e., those electrodes which contributed to global representations (based on a jackknife procedure; see ref. [30]. Our interpretation for this effect is that high levels of gamma power in the positively contributing ("informative") electrodes obscure item-specific spatial patterns and thereby reduce reinstatement of information from encoding. However, more studies are necessary to elucidate the relationship between the magnitude of neural activity measured at various different levels of brain organization (from spike rates to local field potentials and frequency-resolved iEEG data to BOLD responses) to the representation of stimulus-specific information.

Interestingly, only stimulus-specific activity relatively late post stimulus (500–1200 ms) was related to subsequent memory, but not stimulus-specific activity from an earlier cluster (100–500 ms, Fig. 2a). The functional relevance of the later encoding time window is in line with a recent study showing memory-selective neurons responding around 550 ms after stimulus onset[37]. We interpret this result in a levels of processing framework[38] and suggest that activity related to "deeper", more semantic processing stages needs to reoccur during retrieval to support item recognition, while reoccurrence of stimulus-specific activity from a "superficial", more perceptual processing step is not beneficial. This assumption is supported by findings showing that electrodes located in fusiform and inferior temporal lobe are more involved in stimulus-specific representations of early encoding activities while the medial temporal lobe (MTL) seems to be more engaged during representations of late encoding activity. Regions contributing to the early encoding cluster were substantially overlapping with the ventral visual stream[39] that is predominantly relevant for processing perceptual information, while the MTL is a key region for high level information processing and memory[40]. Similarly, only ripple-triggered replay of activity from the late but not the early cluster during nREM sleep supported later memory, suggesting similar benefits of deep representations during ripple-triggered replay.

Contrary to the parsimonious idea that remembered items are replayed while forgotten items are not, we found that the encoding traces of both remembered and forgotten items are spontaneously replayed during the post-encoding rest periods (Fig. 3). We did not find any difference in spontaneous replay levels between remembered and forgotten items. These data suggest that aspects of both remembered and forgotten items are replayed to some extent. However, replay patterns differed between remembered and forgotten items when replay was specifically evaluated triggered to ripples (Fig. 4). For remembered

items, ripple-triggered replay mainly concerned brain patterns from late encoding periods; for forgotten items, neural activity from early encoding time periods was replayed. These results may account for findings that information which only undergoes superficial, but not deep processing cannot be explicitly remembered. This information may still be accessible in implicit memory tests[41], although the relevance of replay for implicit memory needs to be studied further. Previous studies investigating replay of stimulus-specific activity via fMRI observed higher replay of remembered than forgotten items[14,15]; however, due to the relatively low temporal resolution of fMRI recordings, these studies could not distinguish between activities from different processing steps during encoding.

A previous iEEG study investigated replay of distributed sequences of high-gamma activity during sleep[42]. This study reported that replay could indeed be detected (i.e., it was enhanced during sleep periods following as compared to preceding the initial occurrence of these sequences) and that it was related to the timing of several sleep graphoelements including sleep spindles and hippocampal ripples. However, this previous study extracted activity sequences across waking periods characterized by various everyday activity and cognitive experiments, but not locked to specific events. By contrast, we identified stimulus-specific neural representations during a recognition memory task. This allowed us not only to track the re-occurrence of neural activity patterns across multiple processing stages of individual events, but also to relate replay to later memory of individual items and thereby to clarify the functional relevance of replay for memory consolidation.

Another related study[43] provided evidence that local field potential responses recorded via tetrodes in rat parietal cortex are tuned to specific egocentric movements (specifically, to specific combinations of angular and linear velocity), that these responses are reactivated during post-experience sleep periods and that reactivation is linked to hippocampal ripples. These results demonstrate that replay does not only occur at the level of single cells but also on a population level, conceptually similar to the results reported in our current study. This study differs from our study not only in terms of methodology (rats vs. humans, distributed iEEG representations vs. "modules" in parietal cortex, assessment of reactivation etc.), but also with regard to the fact that our design allowed us to assess the functional relevance of reactivation for subsequent memory. Nevertheless, we feel that the studies by Jiang et al.[42] and Wilber et al.[43] (and the results presented here) constitute an important step towards an integration of animal and human research, which should be followed by future investigations in both rodents and humans with parallel experimental designs, recording methodology and analysis methods.

Our study provides first evidence for a differential contribution of spontaneous and ripple-triggered replay to memory consolidation of stimulus-specific representations in humans. These data may explain an apparent paradox of previous studies providing evidence for consolidation-related replay during waking rest[14,15], despite the well-documented beneficial role of sleep for memory consolidation[18–21]. Our data show pronounced differences of ripple-triggered replay during waking rest and nREM sleep, suggesting ripple-triggered replay as a mechanism to account for the beneficial role of sleep for memory.

## Methods

**Epilepsy patients**. Twelve epilepsy patients (8 female; mean age ± standard deviation (SD): 37.9 ± 9.6) participated in the study. No seizure occurred during the test session or during rest. All patients were implanted with invasive EEG electrodes for diagnostic purpose. Medial temporal depth electrodes (AD-Tech, Racine, WI, USA) with ten cylindrical platinum-iridium (diameter: 1.3 mm) contacts were

implanted in all patients. Moreover, subdural grid and strip electrocorticography (ECoG) electrodes with stainless-steel contacts (diameter: 4 mm) were implanted in the frontal, temporal and parietal lobe of several patients. We excluded all electrodes that were ipsilateral to the seizure onset zone. In addition, we excluded contralateral electrodes if they were severely contaminated by epileptiform activity. See Supplementary Fig. 1 for an overview of all remaining implanted electrode contacts that were used for final analyses. Recordings were performed using a Stellate recording system (Stellate GmbH, Munich, Germany) at the Department of Epileptology, University of Bonn, Germany. The study was conducted according to the latest version of the Declaration of Helsinki and approved by the ethics committee of the Medical Faculty of the University of Bonn, and all patients provided written informed consent.

**Experimental design**. The experimental paradigm is depicted in Fig. 1a. The experiment consisted of two consecutive days, a "nap day" and a "no-nap day". On the nap day, patients first learned a set of 80 pictures (remote condition). Half of the pictures were landscapes and the other half buildings. Patients were asked to distinguish between landscapes and buildings by pressing one of two mouse buttons. Afterward, patients were settled in a bed with attenuated light and sound for around 1 h and asked to try to fall asleep. During this period, we conducted polysomnography recordings including scalp EEG, electrocardiograms, facial electromyograms, and recordings of horizontal and vertical eye movements. Fifteen minutes after awakening, patients learned a different set of 80 pictures, again consisting of landscapes and buildings (recent condition), and were again asked to indicate whether they saw a landscape or a building. The recent items served as a baseline, because they cannot be replayed during the nap. After the recent session, the experimenter had a conversation with the patient for around 15 min to avoid rehearsal of items in short-term memory. In the following retrieval session, patients were presented with all pictures from the remote and the recent session, randomly intermixed with 80 new items (again half landscapes and half buildings). They were asked to indicate for each picture if it had been presented before by pressing a left or right mouse button. On the day without nap, the procedure was similar as on the nap day except that patients did not sleep between the remote and the recent session. Otherwise, they could freely choose their behavior during this period. The sequence of the day with and the day without nap was counterbalanced across patients. Continuous EEG data were recorded from the depth and ECoG electrodes during remote encoding, recent encoding, retrieval, and nap periods. A design with a nap day and a day without nap was chosen to investigate the role of an afternoon nap on memory consolidation. Results from this original study in a smaller group of patients than the current study have been reported in previous papers[24,44]. We presented stimuli of landscapes and buildings. Stimuli from two different categories were used in order to ensure attentive processing of the images by requesting subjects to indicate stimulus category by button presses.

**Recording and analyses**. iEEG was referenced to bilateral mastoid electrodes, recorded at a sampling rate of 1000 Hz, and bandpass filtered (0.01 Hz [6 dB/octave] to 300 Hz [12 dB/octave]). All data analyses were based on electrodes from the hemisphere contralateral to the epileptogenic focus to minimize the possibility of artifact contamination. The locations of electrode contacts were ascertained by post-implantation magnetic resonance imaging (MRI). Contacts were mapped by co-registering pre-implantation to post-implantation MRI space, normalizing the co-registered pre-implantation MRI to MNI space and applying the normalization matrix to the post-implantation MRI. The anatomical locations of contacts were then identified using PyLocator ([http://pylocator.thorstenkranz.de/]). All analyses were conducted using the Fieldtrip toolbox[45] and Matlab.

**Extracting item-specific brain patterns during encoding**. In order to optimize the representational specificity of individual electrodes, we first re-referenced all iEEG data by the average activity across all depth and subdural electrodes. Then, we extracted activity from 1 s before the onset of each picture to 1 s after the offset of each picture. Relatively long segments were chosen to remove edge effects associated with spectro-temporal decomposition. EEG trials were visually inspected for artifacts (e.g., epileptiform spikes), and trials with artifacts were excluded from further analyses, resulting in exclusion of 14.22 ± 13.05% (mean ± SD) of all trials. Time-frequency transformation was performed in each EEG trial using complex Morlet wavelet transformation (seven cycles), and power values were extracted. Only data from 100 ms before the onset of each picture to 100 ms after the offset of that picture was kept for further analyses (1.4 s for each trial). We z-transformed the EEG power data within each frequency and each channel (i.e., we normalized the power data by first subtracting the average value and then dividing by the standard deviation across all trials). Further data analyses were based on normalized data. Each trial was divided into 13 time windows of 200 ms, overlapping by 100 ms (−100 to 100 ms; 0–200 ms;…; 1100–1300 ms). Then, we averaged the EEG power within each time window and frequency band of interest (gamma, 30–90 Hz; epsilon, 90–150 Hz; Supplementary Fig. 2a,b; Supplementary Note 2)[28].

Stimulus-specific brain patterns were identified by RSA (ref. [17]; Fig. 1c), using a similar approach as in previous iEEG studies[29,30]. In order to maximize statistical power, we combined activity during remembered and forgotten items from both experimental days. First, we calculated a non-parametric Spearman's correlation

between encoding of a picture ($n$ pictures after artifact rejection, with $n \leq 320$, which is the total number of pictures during the remote and recent sessions across both days) and retrieval of either the same picture ($RSA_{same}$) or a different picture ($RSA_{differ}$). Correlations were calculated across electrodes, separately for each time window during encoding and each time window during retrieval. For each frequency band and patient, this yielded a $times_{encoding} \times times_{retrieval} \times n$ similarity matrix of $RSA_{same}$ values (with $times_{encoding} = times_{retrieval} = 13$ being the number of time windows during encoding and retrieval), and a $times_{encoding} \times times_{retrieval} \times n \times (n-1)$ matrix of $RSA_{differ}$ values. Second, the Fisher-Z transformed correlation matrices were averaged across trials for both $RSA_{same}$ and $RSA_{differ}$, separately for each frequency band and patient. This generated two $13 \times 13$ ($times_{encoding} \times times_{retrieval}$) similarity matrices—one for $RSA_{same}$ and one for $RSA_{differ}$—for each frequency band and patient. Third, we compared $RSA_{same}$ and $RSA_{differ}$ matrices using paired $t$ tests across patients within each $times_{encoding}/times_{retrieval}$ bin and each frequency band. Fourth, we also compared $RSA_{same}$ values against 0 using a one sample $t$ test across patients within each $times_{encoding}/times_{retrieval}$ bin and each frequency band. This was done to ensure that $RSA_{same}$ values were reliably larger than zero.

For each frequency band, these values were corrected for multiple comparisons using surrogate-based cluster statistics[46] (for a similar procedure, see ref. [30]). First, we extracted cluster values from the empirical data (i.e., based on the actual difference between $RSA_{same}$ and $RSA_{differ}$ matrices and between $RSA_{same}$ and 0): We thresholded the $t$ value of each $times_{encoding}/times_{retrieval}$ bin such that only time bins with $t$ values corresponding to $p$ values smaller than 0.01 in both comparisons were taken into account. Time bins with shared edges were defined as belonging to the same cluster. $T$ values ($RSA_{same}$ vs. $RSA_{differ}$) of each contiguous cluster of $times_{encoding} \times times_{retrieval}$ values were summed up. These empirical cluster values were compared to a distribution of surrogate clusters. To obtain these surrogate clusters, we randomly shuffled trial labels (within each patient and frequency band) and performed the same RSA procedure as described above. We extracted clusters from these surrogate matrices and selected the cluster with the largest summary $t$ value ($RSA_{same}$ vs. $RSA_{differ}$) for each surrogate. This surrogate procedure was repeated 10,000 times and produced 10,000 surrogate clusters (if no cluster was found in one of the surrogates, a cluster value of 0 was assumed for this surrogate). Then, we ranked each empirical cluster within the distribution of all surrogate clusters. Clusters with summary $t$ values larger than 99% (alpha level = 0.01) of all surrogate t-clusters were selected. The cluster-based surrogate method effectively controls the alpha level for multiple comparisons on an assumption-free basis regarding the sampling distribution under the null hypothesis[46]. As this analysis was done independently in each frequency band, we additionally Bonferroni-corrected the resulting rank statistics for the two different frequency bands. We also utilized another method to extract surrogate clusters in which we randomly switched condition labels between $RSA_{same}$ and $RSA_{differ}$ at the subject level (i.e., across trial averages). This method generated the same results (i.e., $p < 0.001$).

Next, we compared encoding-retrieval similarity of remembered and forgotten items. In the cluster resulting from the analysis above (i.e., showing significantly higher $RSA_{same}$ than $RSA_{differ}$ values regardless of memory), we extracted $RSA_{same}$ values for remembered and forgotten items, and averaged them separately. This yielded one $RSA_{same}$ value for all remembered items and one for all forgotten items per patient, which were then compared by a paired $t$ test across patients.

**Brain regions supporting stimulus-specific representations.** We conducted a jackknife analysis to evaluate the contribution of different brain regions to stimulus-specific representations within the temporal clusters obtained from the analysis above. This was done separately for each electrode (i.e., at the single subject level) because of the different implantation schemes across patients.

First, we calculated the overall representational similarity indices of each cluster by averaging Fisher-Z transformed correlation values across all time bins within each cluster for each stimulus ($RSA_{total}$). Second, we calculated the correlation between encoding and retrieval by leaving one electrode out ($E[i]$). Again, we calculated the representational similarity indices of $E[i]$ by averaging Fisher-Z transformed correlation values across all bins within each cluster for each stimulus ($RSA_{E[i]}$). We calculated the contribution of each electrode by performing a paired $t$ test between $RSA_{total}$ and $RSA_{E[i]}$ across all stimuli. The higher the difference between $RSA_{total}$ and $RSA_{E[i]}$, the higher was the contribution of $E[i]$.

**Analysis of spontaneous replay during rest.** Patients had on average 1 h and 16 ± 20 min (mean ± STD) of rest. Sleep staging was performed according to the criteria of the American Academy of Sleep Medicine[47] for non-overlapping time windows of 20 s. Across the group of 12 patients, two did not fall asleep during the rest period; ten reached shallow sleep stages (sleep stages N1 and N2); seven reached deep sleep (sleep stage N3, besides N1 and N2); and five had rapid-eye-movement sleep (REM). The percentage of time that patients spent at the different sleep stages is shown in Supplementary Fig. 3. The EEG data during rest was visually inspected for artifacts (e.g., epileptiform spikes), and periods from 2 s before to 2 s after each artifact window were excluded from further analyses. This resulted in exclusion of 6.86 ± 4.22% (mean ± SD) of all data. Time-frequency transformation was performed in each channel using complex Morlet wavelet transformation (seven cycles), and power values were extracted. We then

normalized EEG power using Z transformation (i.e., we first subtracted the average value from the EEG data and then divided by the standard deviation across all time bins within each frequency and electrode).

Replay of stimulus-specific representations was investigated separately for each frequency band during the nap day. We correlated activity during various encoding time windows of each item with activity during each rest time window of each frequency band (Fig. 3a). Activity during encoding was extracted from the time and frequency window showing subsequent memory-related stimulus-specific representations in the analysis described above: Eight time windows centered at time points from 500 to 1200 ms post stimulus (400–600 ms; 500–700 ms;…; 1100–1300 ms) in the gamma frequency range (30–90 Hz; Fig. 2a). Correlations with activity during rest were calculated separately for each remote item (presented before the nap) and each recent item (shown after the nap). We segmented the normalized EEG data during rest into small windows again each lasting 200 ms, overlapping by 100 ms, and averaged EEG gamma (30–90 Hz) power within each window and channel. This results in a specific brain pattern across channels of each rest window. Next, we calculated a Spearman's correlation between the brain pattern of each of the selected encoding time windows of each picture and the brain pattern of each rest window across channels. Spontaneous replay was quantified by averaging all Fisher-Z transformed correlation coefficients across all rest windows, all stimulus-specific time windows and all items, separately for the remote and the recent condition. Finally we performed a paired $t$ test between replay of items shown before the nap (remote) and those shown after the nap (recent) across patients.

In addition, we extracted encoding activity and time windows showing stimulus-specific representations that were not related to subsequent memory: the five gamma power time windows centered at time points from 100 to 500 ms post stimulus (0–200 ms; 100–300 ms;…; 400–600 ms), five epsilon power time windows centered at time points from 300 to 700 ms post stimulus and five epsilon power time windows centered at time points from 500 to 900 ms post stimulus . These results are reported in the Supplementary Notes 5 and 7.

We also investigated the spontaneous replay of the sequence—i.e., the temporal evolution—of activities within the cluster (500–1200 ms) of gamma band activity showing subsequent memory-related stimulus-specific representations during encoding. This was calculated as indicated in Supplementary Fig. 5a: First, we extracted the power time courses in the gamma frequency range (30–90 Hz) within the late encoding time windows (500–1200 ms; 701 data points) and concatenated them across all electrodes. This resulted in a vector with the length of $701 \times n$ ($n$ is the number of electrodes) ($V_{encoding}$) for each item. Second, we segmented the time courses of gamma actvity (30-90Hz) during rest into small windows again each lasting 700 ms (701 data points) and concatenated the gamma time courses across all electrodes. This resulted in a vector with the length of $701 \times n$ ($n$ is the number of electrodes) ($V_{rest}$). We generated a series of $V_{rest}$ patterns by shifting them in steps of 1 ms. Third, we calculated the correlation between $V_{encoding}$ of each item and each of the $V_{rest}$ patterns using Spearman's correlations and Fisher-Z transformed the resulting correlation values. Spontaneous replay was defined as the averaged correlation values, separately for remote and recent items. Finally we performed a paired $t$ test between replay of remote and recent items across patients. We also conducted the corresponding analyses to activity from the early encoding cluster (100–500 ms) for the gamma frequency range, and two clusters (300–700 and 500–900 ms) for the epsilon frequency range. The results are reported in the Supplementary Notes 6, 8 and in Supplementary Fig. 5b–e.

Encoding-time resolved replay was evaluated by correlating EEG power of each encoding time window (from 0 to 1200 ms) with each rest time window separately for the gamma range and the epsilon range. Fisher-Z transformed correlations were averaged across all rest windows and items, separately for each encoding time window of the remote and recent conditions. Finally, we performed a paired $t$ test between the remote and recent condition at each encoding time window across patients within each frequency range. Results were corrected for multiple comparisons (the individual encoding time windows) using an analogous cluster-based procedure as for the analysis of encoding-retrieval similarity. In detail, we first thresholded the $t$ value of each encoding time bin such that only time bins with $t$ values corresponding to $p$ values smaller than 0.05 were taken into account. Adjacent time bins were defined as belonging to the same cluster. $T$ values of each contiguous cluster were summed up. These empirical cluster values were compared to a distribution of surrogate clusters to correct for multiple comparisons. Surrogate data were generated by randomly switching condition labels (remote vs. recent), and performing the same paired $t$ test as applied to the empirical data. We extracted clusters from these surrogate matrices and selected the cluster with the largest summary $t$ value for each surrogate. This surrogate procedure was repeated 10,000 times and produced 10,000 surrogate clusters (if no cluster was found in one of the surrogates, a cluster value of 0 was assumed for this surrogate). Then, we ranked each empirical cluster within the distribution of all surrogate clusters.

**Analysis of ripple-triggered replay during rest.** Ripple events were extracted using a two-step procedure. The first step consisted of a combined manual and automatic artifact detection to reject epilepsy-related high-frequency events. The second step was the ripple extraction itself. First, we selected the hippocampal electrode contact with the least epileptic contamination by visual inspection in each patient and manually rejected larger artifacts (e.g., epileptiform spikes). Automatic

artifact detection was then performed on artifact-free (by visual inspection) data from the hippocampal channel of each patient using a similar procedure as in ref. [27]. We extracted z-scores of the EEG amplitude, of its gradient, and of the EEG amplitude after applying a 250 Hz highpass filter. A time bin was defined as containing artifacts if any of these three indices had a z-score larger than 5, or if any two of these three indices had z-scores larger than 3.

Ripple events were extracted using exactly the same procedure as in a previous study[27]. First, resting EEG data from the selected hippocampal channel were bandpass filtered at the frequency of human ripples (80–100 Hz; we also calculated the 80–140 Hz bands as ripples[24], and found 94.6 ± 4.4% of 80–140 Hz ripples were identified as 80–100 Hz ripples), excluding periods with artifacts (both visually and automatically detected artifacts). Second, the root mean square (RMS) signal was calculated by averaging the filtered EEG data using a moving average of 20 ms. Epochs with RMS values ranking higher than 99% within the RMS values of the entire rest period (including both, nREM sleep and waking state) and lasting more than 38 ms (around three cycles at the lowest ripple frequency of 80 Hz) were marked. Following ref. [27], we only included events that showed at least three peaks or troughs in the unfiltered raw data. These periods together with activity 10 ms before and afterwards (to take temporal smearing of ripple power into account) were defined as "ripple events" (Fig. 4c).

Next, we analyzed replay during ripples and peri-ripple events, using an analogous procedure as for the analysis of spontaneous replay (Fig. 4c). This analysis was conducted separately for ripples occurring during nREM sleep and waking state (Supplementary Note 13). Because only 5 out of 12 patients reached REM sleep and because of the overall short duration of that sleep stage (Supplementary Fig. 3a), we did not analyze ripple-related replay during REM sleep.

First, we extracted the EEG power from ten overlapping 200 ms time windows surrounding each ripple event in the frequency range where stimulus-specific activity had been found (gamma: 30–90 Hz), using seven-cycle Morlet wavelets. EEG power was z-transformed separately for each frequency within the gamma band, across all these extracted time periods in each channel and each patient, and then averaged across all frequencies. We extracted five time windows of 200 ms, overlapping by 100 ms, before and after ripple events, and defined them as "peri-ripple periods" (before ripples: −600 ms with regard to the beginning of the ripple events to −400 ms before; −500 ms to −300 ms; …; −200 ms to the beginning of the ripple event; after ripples: from the end of the ripple event to 200 ms afterwards; 100 ms to 300 ms; …; 400 ms to 600 ms; see Fig. 4c). Furthermore, EEG power in the gamma range (30–90 Hz) was extracted during the ripple events. Second, we calculated the Spearman's correlation between brain patterns of each encoding time window of each item and brain patterns of the ripple event and each peri-ripple period (one correlation with activity during the "ripple event", and ten correlations with time windows during the "peri-ripple period").

Analogous to the analysis of spontaneous replay, data were analyzed by both averaging the (Fisher-Z transformed) correlations during the late (500–1200 ms) and early (100–500 ms) encoding clusters showing stimulus-specific representations, and separately for all encoding time windows (0–1200 ms) in the gamma band. For comparison, the same analysis was then conducted for later forgotten remote items, as well as later remembered and forgotten recent items (Supplementary Note 14).

For both the late and the early encoding cluster in the gamma band, we performed a paired t test between replay levels during ripple events and the averaged replay levels across all peri-ripple periods. Whenever the replay level differed between ripple events and overall peri-ripple periods, we performed an ANOVA and (when significant) post hoc t tests to test if indeed replay during each individual time window of the peri-ripple period differed from replay during the ripples.

For the individual analysis of all encoding time windows (0–1200 ms), we again compared replay levels during ripple events and averaged replay levels across all peri-ripple periods. Results were corrected for multiple comparisons (the individual encoding time windows) using an analogous cluster-based procedure as described above. Surrogate data was generated by randomly switching condition labels (ripple events vs. peri-ripple periods).

## Data availability

All relevant data are available from the authors upon reasonable request, including the iEEG data, behavioral log files, pre-implantation MRI, post-implantation MRI, and MATLAB scripts. Patients' private identifications are all anonymized.

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

## Acknowledgements

H.Z. was supported by SFB 874. J.F. was funded via SFB 1089 and DFG grant FE366/9-1. N.A. received support via DFG (SFB 1280/A02, A03, F02; SFB 874/A11, B11; and project AX82/3).

## Author contributions

H.Z., J.F. and N.A. designed the experiment. H.Z. analyzed the data. H.Z. and N.A. wrote the manuscript.

## Additional information

**Competing interests:** The authors declare no competing interests.

