## [Peer Review File · Nature Communications]

Reviewers' comments:

Reviewer #1 (Remarks to the Author):

In the present manuscript, Zhang et al. investigate a putative mechanism of memory consolidation, namely spontaneous replay. Replay has first been described in rodents and since then numerous studies in humans attempted to provide evidence for a similar mechanism. However, most human studies utilized non-invasive imaging (e.g. fMRI). Here Zhang et al. take advantage of intracranial recordings in human epilepsy patients to tackle that question. They report that neural representations during encoding are stronger for remembered than forgotten items (Fig 2) and then demonstrate increased spontaneous replay levels for remote vs. recent items, which were independent of behavioral outcome (remembered vs. forgotten) and did not differ between wake and NREM sleep (Fig. 3). However, despite this null result (no specific role of sleep), they then assessed whether replay is associated with ripple oscillations in the hippocampus and found evidence for this consideration. In Fig 4 they report that replay is apparently increased during the ripple, but this seems to have to specific behavioral relevance. In addition, they provide substantial supplemental information. In general, the paper is well written and well structured.

1. The results appear incremental to previous fMRI studies. While the authors use intracranial EEG, which provides a superior spatiotemporal resolution, they then limit the analyses to broadband gamma activity and analyze the data in 100ms time windows, which greatly reduces the temporal resolution.
2. While the authors provide several detailed analyses, it seems that (a) replay is not specific to ripples, (b) replay and ripples are not specific to sleep and (c) does not have a clear behavioral benefit. Given that their abstract heavily features the importance of NREM sleep, the key question how NREM sleep subserves memory consolidation remains unaddressed.
3. The choice of frequency bands is rather unfortunate. Why did the authors artificially split the frequency bands into gamma (<90) and epsilon (>90), when the putative replay-triggering event is at 80-100 Hz? In addition, Fig S2 indicates that there is no clear separation between the two bands, so why was this split necessary? It seems the authors used this step to render some results significant (e.g. Fig 2), which otherwise wouldn't have turned out significant.
4. Some key data in Fig2-4 is presented as bar plots and should be superimposed with single subject scatter plots to assess how these effects might generalize across subjects.
5. Did the authors consider autocorrelations in the signal that might affect their RSA estimates for recent vs. remote events?
6. Statistical testing in fig 3e is unclear – if testing was carried out on 3e.i (corrected for multiple comparisons? multiple t-tests or ANOVA with factor time bin?), then 3e.ii constitutes double dipping if the authors collapsed across the 3 significant time periods and tested again for significance
7. Fig4. It is unclear where one should appreciate the effect that the authors describe? The bar plots are not informative. What is on the y-axis of Fig 4d and Fig4e and how does it relate to Fig 2/3. In particular, since the absolute values seems higher than in Figure 3. Would that indicate a stronger reactivation during sleep? What does Figure 4c show? Especially on the left side, it seems that one only notices increased replay due to increased gamma power correlations due to the ripple. As a control, the authors could test replay in states of elevated gamma power that are not classified as ripples.
8. Ripple specificity: The authors need to demonstrate that replay is not only enhanced in the peri-ripple time window (before and after), but also compare it to random epochs that are not being selected based on increased high gamma power.
9. How do RSA values change with? Is there a general trend in the data that is independent of wake vs. sleep?
10. Did the authors consider time-compressed replay?

Taken together, the authors present an impressive list of analyses. Some findings are interesting to broad range of neuroscientists (e.g. behavioral relevance during encoding in Fig. 2), but there is

only little evidence for ripple-triggered replay. In addition, it seems that the described effects are not specific to NREM sleep, but also occur during wakefulness. A major caveat is that replay as studied here does not predict subsequent behavioral outcome. However, this paper adds to the existing literature and e.g. describes differential replay effects of remembered and forgotten items in terms of their relative timing to each other (Fig 4d).

Reviewer #2 (Remarks to the Author):

In this study, Zhang et. al. use intracranial recordings from patients with epilepsy and identify stimulus elicited changes in gamma power which tend to replay during post encoding sleep and wakeful rest. They report high levels of spontaneous replay of neural activity, measured during the early and late encoding phases, in sleep and rest. Levels of spontaneous replay were not different when comparing the later remembered trials versus the forgotten ones in the post sleep retrieval task. The neural representations of remembered and forgotten items have higher levels of replay of their late and early encoding phases respectively, during ripple events compared to peri-ripple windows. This is only seen in nREM and not waking state, further supporting the idea that sleep is essential for memory consolidation.

Critique

This paper tackled questions that can elucidate mechanisms of memory replay and consolidation in humans. It provides electrophysiological evidence for the occurrence of spontaneous and ripple-triggered replay of stimulus-elicited changes in gamma power. The authors parsed out the difference in replay of the early vs. late time durations of stimulus locked responses and relate those differences to memory performance. They also differentiated between replay during sleep and wakeful rest, and distinguished between both spontaneous and ripple-triggered replay. They used appropriate controls and addressed some confounds. However, there are few important points that need to be addressed: 1) The paper lacks discussion on the spatial dynamics of replay of stimulus specific representation. The paper could have identified the functional regions, which contributed most to their findings both at the group and single subject level. 2) The paper doesn't address a potential confound regarding the number of ripples detected in or the duration of recording of wakeful state which may explain why their results don't show that awake replays are related to memory performance. 3) This study is not (as claimed) the first electrophysiological evidence for replay or ripple-triggered replay in humans (see Jiang et al., *Replay of large-scale spatio-temporal patterns from waking during subsequent NREM sleep in human cortex*. *Sci Reports* 2017). 4) This paper could potentially contribute to linking extensive electrophysiological reactivation studies in rodents and fmri studies in humans. However, the link is not straightforward due to some critical differences in experimental design, such as the absence of pre-behavior sleep baseline in the present study. 5) There were many analyses that were not cohesively described in the methods section, and several figures that did not have sufficient caption information. Please see suggestions below on how to address these points.

Major comments

1. The method described on Page 12 (first paragraph) is similar to template matching method for detecting replay in rodent literature (e.g. Louie and Wilson, 2001), but the results would be more convincing if the replay levels for real gamma temporal pattern were compared to the distribution of replay levels for surrogate patterns (e.g. constructed by shuffling the pattern time bins). In addition, replay in rodent studies is typically time compressed and it would be interesting address this question in the present study.
2. What was the spatial distribution of electrodes for each patient? Authors should try and remove electrodes from each region at a time, see if significance still holds, and assess which electrodes contribute most to their results. It would be informative to know if there is a spatial pattern.
3. [Page 7 third paragraph line 7] The authors discussed that the early encoding cluster may play a role in more superficial, perceptual processing, and the late encoding cluster may play a role in

deeper, more semantic processing. To provide better evidence for this, the authors can consider whether lower levels sensory areas such as the occipital or occipito-temporal electrodes contribute more to the early encoding gamma trace while higher level cortical association areas such as VTC, frontal, and parietal contribute more significantly to the later encoding phase.

4. [3 Page 11 Supplementary figure] The authors' finding that replay in awake state is not related to memory performance is contrary to the rodent literature and the authors did not address this issue in sufficient detail. The limited number of ripples occurred during wakeful rest can explain their lack of finding. The authors should mention if sleep and wakeful rest durations were matched. The hypnograms presented in supplementary materials show that patients spent a much longer duration in sleep than in awake state. It is probable that there was insufficient wakeful rest data. The brain is also more dynamic during awake state and much longer durations may be necessary to ensure that the brain spends enough time in the state in which ripples occur.

5. Rodent electrophysiology replay experiments typically use pre-behavior sleep as a baseline for comparison with post-behavior sleep. However, in this study, the authors compared replay of pre-sleep task with sleep, and that of post-sleep task with sleep. The experimental design in this study is different from the traditional approach and the authors should discuss their reasons for this experimental design.

6. Authors are citing studies showing higher BOLD signal in areas involved in recent learning as evidence for replay in humans (e.g. Peigneux et al., 2004), and present their results as an electrophysiological demonstration of the same phenomena. Although BOLD signal is positively correlated with gamma power (Logothetis et al., 2001), replay scores and gamma power in their results are slightly negatively correlated. This discrepancy requires discussion.

7. In the terms of novelty, this is not the first electrophysiological evidence of replay or ripple-triggered replay in humans (see Jiang et al., Replay of large-scale spatio-temporal patterns from waking during subsequent NREM sleep in human cortex. *Sci Reports* 2017), nor the first demonstration of replay using population activity reflected in LFP signal (see Wilber et al., Laminar organization of encoding and memory reactivation in the parietal cortex. *Neuron* 2017). These studies should be cited and the strong novelty claim ("Our study, for the first time in humans, distinguished spontaneous replay from ripple-triggered replay") should be revised accordingly.

8. [Page 20 figure 4C] Figure 4C shows results specific for the remembered and forgotten trials. These trials consist of two types of stimuli, landscapes and buildings. It should be mentioned if the number of stimuli from each category is matched across forgotten and remembered trials as this may be a confound. For example, building trials may be more easily distinguishable by the patient, and therefore the higher performance may not be due to the intra-ripple replay of the later encoding cluster.

a. In addition, it is not mentioned if the images shown are matched for color, size, and intensity. There are many electrodes in the temporal lobe and ventral visual pathway, which respond selectively to certain features of visual stimuli.

9. [Page 20 figure 4C] The authors compared remote (pre-sleep items) spontaneous replay to recent (post sleep items) replay, but not remote ripple triggered replay to that of recent. When comparing the remote spontaneous replay, it was not found to be statistically higher than that of recent in nREM sleep. Therefore, it should be confirmed that the ripple-triggered replay seen in nREM sleep of the remote items is significantly higher than baseline (recent items).

10. The authors discussed the experimental design of nap vs. no nap day, and landscape vs. building stimuli. What is the purpose of using nap versus no nap days, and what is the reason for using two categories of visual stimuli? This should be discussed explicitly in the paper. It would also be interesting to know if patient accuracy was higher during the nap days.

11. How was stimulus presentation duration determined? Did stimulus duration have a fixed interval, or was it based on the patient's response time? If during encoding, the stimulus was still presented after the patient responded, then motor effects need to be accounted for in the neural signal. Additionally, response time should be discussed in relation to the time of elicited neural response. Lastly, please discuss whether patients were using their dominant hand for both button presses during encoding.

12. It would be informative to clarify the size of peri-epileptic discharge window excluded from analysis and what proportion of trials/nap epochs were on average lost to exclusion?

Minor comments

1. The main text and supplementary figures were not documented well enough by captions.
2. There is a typing error in the second paragraph of results in the third line 'a with regard to encoding...'
3. There are several accidental font changes: third to last line on page 3 and last sentence of first paragraph on page 4, and several more
4. Missing space in page 2 line 3
5. Page 4, Line 11 and Page 6, Line 2. P values ~ 0.1 or ~ 0.07 should not be interpreted as indicating significant result, especially for a claim as strong as the higher replay levels of early gamma component predict lower memory performance. In addition, different p values are reported for what seems to be the same comparison ($p = 0.071$ on Page 6, Line 2 and $p = 0.061$ in Fig.4 caption, Page 20, Line 8).
6. Page 10, Line 16 and elsewhere. Matrixes should be matrices.
7. Page 20, Fig.4A. Please note if the plot is centered at ripple onset (as it appears) or peak?
8. Page S12, Fig.S4 caption, Line 5. It seems like 'x' significance label is in the figure caption, but not on any of the plots.
9. Page S12, Fig.S4. For line plots, it would be helpful to use different colors for different lines, as well as shaded error bars.
10. Page S16, Fig.S8A. Time windows with significant difference should be denoted by symbol at the top of the plot, rather than using error bars of different color.

Reviewer #3 (Remarks to the Author):

This manuscript, 'Electrophysiological mechanisms of human memory consolidation,' explores the interesting question as to whether hippocampal ripples as captured by intracranial EEG in patients with medically refractory seizures are linked to replay of neural activity during memory consolidation. This is a highly relevant study, and addresses an important question in the field. Previous evidence has suggested that activity present when items are encoded into memory are replayed. This happens both during retrieval, but also during times of memory consolidation. The authors begin their study here by demonstrating that investigating these replay events during a rest period following study (the rest period can either be when subjects are asleep or awake). The novelty of the manuscript is in demonstrating that these replay events are coincident with hippocampal replay during the rest period. This would be an important contribution, as it links previous rodent work demonstrating the presence of these ripples, and linking that work to human memory consolidation.

This is a good study, and the authors should be commended for both their analyses and their efforts. This would be a valuable addition to the literature, providing direct evidence of the role of hippocampal ripples in replay events. The manuscript and the analysis is a bit complicated, given the many comparisons between encoding time periods, remote and recent items, and awake and asleep states. Nonetheless, the results appear to be well supported. I feel there are a few concerns that should be addressed.

Major concerns

The primary result rests on two major components. First, when examining the entire rest period,

there appear to be greater replay events for remote stimuli compared to recent stimuli (a baseline control) and that replay activity is greater during ripple compared to peri-ripple time periods. Clearly, over the entire hour long rest period, there should only be a few time points during which replay events occur, and so the fact that there are greater levels of replay averaged over this entire period are remarkable given the amount of noise and replay independent activity that would occur during this time. I thought the comparison between ripple and peri-ripple time periods was interesting, but a more direct test to ask whether ripples mediate replay would be to examine whether the replay levels observed over the entire rest period actually return to baseline levels once those ripple time points are removed. In other words, rather than compare ripple time points to only the peri-ripple time periods, the authors should also demonstrate that removing the ripple time points results in a significant reduction in overall replay during rest. Or conversely, compare the replay during ripple events to replay during the remainder of the rest period.

The second major concern I have relates to the memory relevance of these ripple events. Most of the analyses focus on the late encoding period, and suggest that memory relevance is only present when examining activity from 500-1200 ms during encoding, even though the early time periods also demonstrated stimulus specific replay. The authors explain this result by suggesting that there is a deep encoding period that occurs that is relevant for memory formation that is distinct from superficial stimulus processing. However, as it relates to ripples, it appears that the only time periods that show a significant memory related modulation occur even later during encoding, as demonstrated in Figure 4e (around 1200 ms). It is difficult to understand, then, how the ripple related replay events that correspond to earlier time points are relevant for memory, and why such distinctions occur only at the very end of the encoding period.

The authors present an analysis examining whether there is replay of the entire sequence. To do this, they construct larger concatenated matrices corresponding to 8 sequential time points, and examine the correlations of those larger matrices. However, to me, this appears to simply temporally smear the replay data that were analyzed using more temporally precise 200 ms window. In order to examine the sequence of activity, then an analysis should be performed on the trajectory of activity over this time window, rather than just the average correlation over the concatenated window. This is a supplementary point to the main results, in my view, and so unless a specific demonstration of how the activity progresses from time point to time point can be shown, it is hard to understand how this provides evidence of the sequence.

There was no evidence that there was a greater level of replay for remote compared to recent items during nREM sleep. Interestingly, the only differences occurred when comparing remote to recent replay during wake. Yet ripple events in their analysis were focused on nREM sleep. This point requires some clarification. Do ripple events only occur during sleep in their analysis (they do not show the distribution of ripple events during the different states of vigilance)? If so, then how should we interpret the fact that nREM sleep does not contain more replay events for remote compared to recent items.

For the cluster analysis to correct for multiple comparisons, the authors state that they compare the empiric cluster to all clusters derived from each permutation. There are two points that should be clarified. First, the clusters used to derive the null distribution should only be comprised of the maximum cluster statistic from every permutation, not all possible clusters (as an individual permutation may have more than one significant cluster). Second, it is not clear at what level the permutations are being conducted. If comparing across subjects, the permutations should be performed at the level of individual subjects, not the level of individual trials, so that the subject is the unit of observation. This would result in 2^{12} possible permutations (given 12 subjects) when comparing one condition to another, of which they could choose just 10,000 to make their null distribution. It is possible that this permutation was indeed performed at the level of subjects, but this is not clear.

Reviewers' comments:**Reviewer #1 (Remarks to the Author):**

In the present manuscript, Zhang et al. investigate a putative mechanism of memory consolidation, namely spontaneous replay. Replay has first been described in rodents and since then numerous studies in humans attempted to provide evidence for a similar mechanism. However, most human studies utilized non-invasive imaging (e.g. fMRI). Here Zhang et al. take advantage of intracranial recordings in human epilepsy patients to tackle that question. They report that neural representations during encoding are stronger for remembered than forgotten items (Fig 2) and then demonstrate increased spontaneous replay levels for remote vs. recent items, which were independent of behavioral outcome (remembered vs. forgotten) and did not differ between wake and NREM sleep (Fig. 3). However, despite this null result (no specific role of sleep), they then assessed whether replay is associated with ripple oscillations in the hippocampus and found evidence for this consideration. In Fig 4 they report that replay is apparently increased during the ripple, but this seems to have to specific behavioral relevance. In addition, they provide substantial supplemental information. In general, the paper is well written and well structured.

1) The results appear incremental to previous fMRI studies. While the authors use intracranial EEG, which provides a superior spatiotemporal resolution, they then limit the analyses to broadband gamma activity and analyze the data in 100ms time windows, which greatly reduces the temporal resolution.

Response: We are glad to hear that in general the Reviewer appreciates the writing and structure of our manuscript. We indeed used a similar design as previous fMRI studies, i.e. two encoding sessions before and after a nap (e.g.[1]). However, we would like to emphasize several substantial novel contributions of the current manuscript which we feel are not merely incremental:

- (1) Our study is the first to show spontaneous reactivation of stimulus-specific activity patterns in intracranial EEG recordings. While this result is conceptually similar to previous fMRI papers on replay (e.g.,[1, 2]), we could show that replay occurs specifically for activity in the gamma frequency range rather than for lower-frequency activity. This result is consistent with previous studies showing that stimulus-specific gamma-band activity during encoding of an item is reinstated during long-term memory retrieval [3, 4].**
- (2) The superior temporal resolution of iEEG as compared to fMRI allowed us to disentangle replay of activity that was processed during different time windows at encoding. Our findings show that memory consolidation relies on replay of late rather than early encoding activity.**
- (3) Most importantly, we provide first evidence that replay during hippocampal ripple oscillations supports memory consolidation in humans, which obviously cannot be detected with fMRI. Our result that ripples during nREM sleep, but not during awake resting state, promote replay and later memory also allowed us to resolve an apparent paradox in the previous literature: On the one hand, several fMRI studies reported replay not only during sleep but also during awake resting states (e.g., [1, 2]). On the other hand, behavioral studies demonstrated a clear benefit of sleep for memory consolidation [5, 6], and studies using targeted memory reactivation even found that reactivation during waking state impairs later memory [7]. We show that replay occurs during both awake resting state and sleep, whereas ripple-triggered replay is restricted to NREM sleep and is particularly relevant for later memory.**

We restricted our analysis to relatively long time windows in line with previous intracranial EEG studies that employed representational similarity analyses across electrodes to identify stimulus-specific representations: In our previous study [3] we used exactly the same temporal resolution as in the current study (200ms temporal epochs, spaced by 100ms). Studies from other groups using representational similarity analyses applied similar or even longer time windows (500ms temporal epochs spaced by 100ms in [4] and [8]; 200ms temporal epochs spaced by 10ms in [9]). These relatively long time windows were used in order to be able to reliably extract stimulus-specific representations from activity across electrodes.

In fact, selecting time windows of 200ms duration is optimal given the temporal autocorrelations in the gamma frequency band, which shows significant autocorrelations up to 250ms (see the new Figure S2d shown below in response to comment #5). Thus, selecting smaller windows would not add substantial information.

We also performed a control analysis at a slightly higher temporal resolution (100ms temporal epoch, spaced by 50ms). The analysis procedure was the same as for the main analysis: We first extracted for each channel and each item the iEEG activity within the gamma band (30-90Hz), but now using consecutive time windows of 100ms, overlapping by 50ms. The temporal resolution was thus twice as high as in our main data. We then applied the same analysis as shown in Fig. 1c, i.e. analyzing encoding-retrieval similarity of same and different items at different encoding and retrieval time windows. Consistent with our previous results, we observed clusters showing significantly higher correlation between encoding and retrieval of the same items compared with different items. As illustrated in Fig. S2b, three separate clusters were identified: 150ms-550ms, 650ms-1,200ms, and 850-1,000ms. Again, we found that activity during later, but not earlier time windows during encoding is behaviorally relevant for later retrieval, because encoding-retrieval similarity was higher for remembered than forgotten items in the late (650-1,200ms) cluster ($t(11)=2.78$, $p=0.018$), but not the other two clusters (both $t(11)<1.01$, both $p>0.34$).

Please see page 3, lines 5-7; Supplementary Results 2.2 and 2.3; and Fig. S2b,d for additional results.

Page 3, lines 5-7: “we also tried consecutive time window of 100ms, overlapping by 50ms and it generated highly similar results; Fig. S2b; Supplementary Results 2.2”

Supplementary Results 2.2 “Stimulus-specific representations based on gamma activity extracted with a high temporal resolution

We first extracted for each channel and each item the iEEG activity within the gamma band (30-90Hz) and used consecutive time windows of 100ms, overlapping by 50ms. The temporal resolution was twice as high as the one reported in our main data. We then applied the same analysis as described in Fig. 1c, i.e. analyzing encoding-retrieval similarity of same and different items in different time windows. Consistent with the results reported in the main paper, we observed clusters showing significantly higher correlation between encoding and retrieval of the same items compared with different items. As illustrated in Fig. S2b, three separate clusters were identified: 650ms-1,200ms, 150ms-550ms, and 850-1,000ms. Again, we found that activity during later, but not earlier time windows during encoding was behaviorally relevant for later retrieval,

because encoding-retrieval similarity was higher for remembered than forgotten items in the late (650-1,200ms) cluster ($t(11)=2.78$, $p=0.018$) but not the other two clusters ($t(11)<1.01$, $p>0.34$). This result is highly consistent with the results reported in the main paper, which were based on consecutive time window of 200ms, overlapping by 100ms.”

Supplementary Results 2.3 “Autocorrelation effects

We estimated the amount of temporal autocorrelations in distributed representations at gamma power (30-90Hz). First, we correlated 30-90Hz EEG power between any two time bins of each encoding trial across electrodes (using time bins of 10ms). This analysis results in an $n \times n \times m$ matrix for each participant (n : number of time bins in each trial; m : number of trials). Second, Fisher-Z-transformed correlation values were averaged across trials by matching the temporal distance between two correlated time bins. Correlations corresponding to a temporal distance of 0 were discarded. This generated a $l \times (n-1)$ vector for each participant. Finally, we compared the correlation value at each temporal distance with the average correlation value across all temporal distances by performing a paired T-test across participants. We found that the autocorrelation was significantly above the average only during intervals of up to 250 milliseconds (Fig. S2d). Autocorrelations between bins with intervals above 250 milliseconds did not differ from the averaged value. This result is consistent with results from a similar analysis in our previous study [3] showing that the temporal autocorrelation of intracranial EEG data decays quickly, in particular at high frequencies. Since autocorrelations are restricted to such short time intervals, they do not affect correlations between encoding and retrieval or between encoding and resting state.”

Fig. R1 (Fig. S2b,d). *b) Three clusters showed stimulus-specific representations (i.e., higher correlations between encoding of one and retrieval of the same item as compared to encoding of one and retrieval of a different item) in the gamma range using a high temporal resolution with consecutive time window of 100ms, overlapping by 50ms. P-values beside each cluster indicate the results of assessing the functional relevance of*

*stimulus-specific representations for memory (comparing RSA values between remembered and forgotten items within each cluster). d) Assessment of autocorrelations. The right panel is a magnification of the left panel. Red *s indicate correlations higher than the averaged correlations. The green line indicates the averaged correlation values.*

2). While the authors provide several detailed analyses, it seems that (a) replay is not specific to ripples, (b) replay and ripples are not specific to sleep and (c) does not have a clear behavioral benefit. Given that their abstract heavily features the importance of NREM sleep, the key question how NREM sleep subserves memory consolidation remains unaddressed.

Response: We agree that the mechanism whereby nREM sleep benefits memory consolidation is a key question. We also generally agree with the three statements made by the Reviewer but would like to add some clarifications which, in our opinion, indeed constitute a substantial step forward in understanding how NREM sleep subserves memory consolidation:

- (a) *“replay is not specific to ripples”*: Indeed our findings of spontaneous replay (Fig. 3) as well as previous studies [1, 2] indicate that replay of stimulus-specific information does not only occur during ripples, but also spontaneously in the absence of ripples. Nevertheless, we do find that replay is increased during ripples as compared to resting periods before and after ripples (Figure 4) and as compared to randomly selected resting periods (see below, comment #10). Furthermore, replay was higher during ripples than during periods showing enhanced power within the ripple frequency band (80-100Hz) in non-hippocampal electrodes (“surrogate ripple events”, see below, comment #9). In addition, we found that replay during ripples differs in important respects from spontaneous replay: while spontaneous replay occurs during both, nREM sleep and waking state, and is not related to later memory for an item (Figure 2), ripple-triggered replay is specific to nREM sleep and differs between subsequently remembered and forgotten items. This suggests that ripple-triggered replay differs from spontaneous replay both quantitatively (related to the “strength” of replay) and qualitatively (related to the state-dependence and functional relevance of replay).
- (b) *“replay and ripples are not specific to sleep”*: Again, we agree that replay occurs during both, awake resting state and nREM sleep. However, as described above, ripple-triggered replay during nREM sleep differs substantially from spontaneous replay during waking state. Ripples were found to occur during both resting state and sleep, consistent with previous observations in both rodents [10, 11] and humans [12-14]. However, they support later memory specifically during nREM sleep.
- (c) *“replay does not have a clear behavioral benefit”*: As just described, our current results indicate differential roles of spontaneous and ripple-triggered replay and suggest that ripple-triggered replay (if it occurs during nREM sleep) supports memory consolidation, while spontaneous replay does not.

Indeed, we feel that the previous version of the Abstract does not report these points with a sufficient degree of precision. In particular, the concluding statement “These results provide the first electrophysiological evidence for replay in humans, and suggest a mechanistic explanation for the beneficial role of sleep in memory consolidation.”

needs to be explained further. We have added the following sentence on page 1, lines 19-20:

“Specifically, they point to a prominent role of ripple-triggered replay during nREM sleep in consolidation processes.”

3). The choice of frequency bands is rather unfortunate. Why did the authors artificially split the frequency bands into gamma (<90) and epsilon (>90), when the putative replay-triggering event is at 80-100 Hz? In addition, Fig S2 indicates that there is no clear separation between the two bands, so why was this split necessary? It seems the authors used this step to render some results significant (e.g. Fig 2), which otherwise wouldn't have turned out significant.

Response: The gamma and epsilon band were defined according to commonly used specifications [15]. The same is true for the ripple band, where we followed the band definition in [16]and [17]. We intended the specifications of the gamma and epsilon band to be independent of the specification of the ripple range to avoid a possible bias (i.e. replay findings in the gamma or epsilon range due to ripple events; see also response to comment 8). Importantly, finding functionally relevant stimulus-specific representations is a necessary prerequisite for, but analytically independent from, the assessment of both spontaneous and ripple-triggered replay. Also, please note that the gamma band underlying stimulus-specific representations was computed across all electrodes, while the ripple frequency band was based on the channels located in the hippocampus.

Nevertheless, in the revised manuscript, we assessed the contribution of different frequency ranges to stimulus-specific representations during encoding in greater detail. We found that functionally relevant stimulus-specific representations occurred only at frequencies up to 90Hz but not above: In detail, during encoding, we investigated stimulus-specific representations within four sub-bands between 30 and 150Hz (30-60Hz, 60-90Hz, 90-120Hz, and 120-150Hz). We applied the same RSA approach as reported in the manuscript for the main analysis. Within each sub-band, we correlated the distributed EEG power between encoding and retrieval of the same item across electrodes and compared it with correlation values between encoding and retrieval of different items. As indicated in Fig. R2 (Fig. S2aii) shown below, we found functionally relevant (i.e., memory-related) stimulus-specific clusters within the 30-60Hz and 60-90Hz bands but not within the sub-bands above 90Hz (90-120Hz and 120-150Hz) (by performing a paired t-test between RSA values of remembered vs. forgotten items across participants within each cluster). Notably, we also found functionally relevant stimulus-specific representations when excluding activity from the ripple band, i.e. when focusing on 30-80Hz activity (see Fig. R2). These results are now added in Supplementary Results 2.1.

Together, these results show that stimulus-specific representations during encoding do not depend on including activity from the ripple-frequency band.

Fig. R2. (Fig. S2a ii): Clusters showing stimulus-specific representations (i.e., higher correlations between encoding of one and retrieval of the same item as compared to encoding of one and retrieval of a different item) for four different sub-bands between 30-150Hz and for a gamma band between 30-80Hz. P-values beside each cluster indicate the results of assessing the functional relevance of stimulus-specific representations for memory (comparing RSA values between remembered and forgotten items within each cluster).

4). Some key data in Fig2-4 is presented as bar plots and should be superimposed with single subject scatter plots to assess how these effects might generalize across subjects.

Response: We thank you for this suggestion and changed the figures accordingly.

5). Did the authors consider autocorrelations in the signal that might affect their RSA estimates for recent vs. remote events?

Response: This is a very important point that we have carefully checked. We have addressed this issue in the Supplementary Results 4.3, where we show that correlation values between items during the two encoding sessions (i.e., remote items and recent items) and the rest period do not depend on temporal distance.

In order to directly quantify autocorrelations of distributed representations in the gamma frequency range, we conducted an additional control analysis which is now reported in the Supplementary Results 2.3 and in Fig. S2d.

In the main text on page 3, line 4 we write: “autocorrelation of the gamma activity was significant up to 250ms; Supplementary Results 2.3; Fig. S2c”

Supplementary Results 2.3, Autocorrelation effects:

“We estimated the amount of temporal autocorrelations in distributed representations at gamma power (30-90Hz). First, we correlated 30-90Hz EEG power between any two time bins of each encoding trial across electrodes (using time bins of 10ms). This analysis results in an $n \times n \times m$ matrix for each participant (n : number of time bins in each trial; m : number of trials). Second, Fisher-Z-transformed correlation values were averaged across trials by matching the temporal distance between two correlated time bins. Correlations corresponding to a temporal distance of 0 were discarded. This generated a $1 \times (n-1)$ vector for each participant. Finally, we compared the correlation value at each

temporal distance with the average correlation value across all temporal distances by performing a paired T-test across participants. We found that the autocorrelation was significantly above the average only during intervals of up to 250 milliseconds (Fig. S2d). Autocorrelations between bins with intervals above 250 milliseconds did not differ from the averaged value. This result is consistent with results from a similar analysis in our previous study [3] showing that the temporal autocorrelation of intracranial EEG data decays quickly, in particular at high frequencies. Since autocorrelations are restricted to such small time intervals, they do not affect correlations between encoding and retrieval or between encoding and resting state.”

Fig R3 (Fig. S2d). Assessment of autocorrelations. The right panel is a magnification of the left panel. Red *s indicate correlations higher than the averaged correlations. The green line indicates the averaged correlation values.

6). Statistical testing in fig 3e is unclear – if testing was carried out on 3e.i (corrected for multiple comparisons? multiple t-tests or ANOVA with factor time bin?), then 3e.ii constitutes double dipping if the authors collapsed across the 3 significant time periods and tested again for significance

Response: We apologize for the lack of clarity. Figure 3e.i shows the results of multiple t-tests of replay of each encoding time bin. These results were corrected for multiple comparisons using surrogate-based cluster statistics – i.e., we summed significant uncorrected t-values across adjacent significant time bins to create cluster values, then randomly shuffled the assignment of values to the two conditions (remote vs. recent) at each time bin. We then extracted for each permutation the surrogate cluster with the largest sum of t-values. This was repeated 10,000 times. Finally, we compared each empirical cluster value to the distribution of surrogate clusters.

The three clusters in Figure 3e.i indicate higher spontaneous replay levels of remote items as compared to recent items. Figure 3e.ii shows the results of a different test, namely the comparison of replay levels of remote items to zero. This analysis thus addressed the question whether the higher replay levels of remote than recent items within the three clusters is due to enhanced replay of remote items or decreased replay of recent items. As illustrated in Fig. 3e.ii, remote replay of the 100-200ms cluster did not differ from zero, but the replay levels of the 500-700ms and 1,100-1,200ms clusters were significantly above zero.

In the revised manuscript, we changed the legend of Fig. 3e as follows:

“(e) Encoding-time resolved replay during rest. (ei) Reactivation differs between remote and recent items in three time periods marked with *s in different colors after cluster-based correction for multiple comparisons (surrogate data was generated by switching remote and recent labels). (eii) Averaged replay levels of all clusters depicted in ei as compared to zero.”

7). Fig 4. It is unclear where one should appreciate the effect that the authors describe? The bar plots are not informative. What is on the y-axis of Fig 4d and Fig4e and how does it relate to Fig 2/3. In particular, since the absolute values seems higher than in Figure 3. Would that indicate a stronger reactivation during sleep? What does Figure 4c show?

Response: Again, we apologize for the confusion. The bar plot in Fig. 4d shows that ripple-triggered replay of early and late encoding activity has a different effect on memory, as reflected by a significant interaction between “cluster” (early vs. late) and “memory” (remembered vs. forgotten). The late and the early cluster are corresponding to the late and early cluster in Fig. 2a. The y-axis shows replay levels (i.e., correlations between encoding activity and rest) as in Figure 3b-e and as color-coded in Figure 4c.

We have changed the scale in Figure 3 (from 10^{-2} to 0.01) in order to facilitate this comparison. We have also modified the legend of Figure 4d. It now reads: “Ripple-triggered replay of early and late encoding activity differentially affects memory: Interaction between replay levels of remembered and forgotten items and gamma band activity from the early (100-500ms) and the late (500-1,200ms) encoding clusters.”

Indeed, the ripple-triggered replay levels in Figure 4d are higher than spontaneous replay levels (extracted during surrogate ripple events as well as randomly selected epochs; see response to comments #9 and #10). This reflects the increased replay during ripples.

Figure 4c depicts replay levels (i.e., correlations between activity during encoding and activity during ripple events and peri-ripple periods) of each encoding time bin (plotted on the vertical axis) during different time bins triggered to ripples (plotted on the horizontal axis) – i.e., during the ripple (time bin “0” on the horizontal axis) and during the peri-ripple periods (time bins before and after the ripple). We changed the axis labels and figure captions of Figure 4 in order to improve clarity.

8). Especially on the left side, it seems that one only notices increased replay due to increased gamma power correlations due to the ripple.

Response: We agree that this is an important point. We have conducted several control analyses to exclude that replay during ripples is due to the increased power during ripples. First, we showed that the effect is specific to ripples occurring during nREM sleep and does not occur during waking state ripples, also when exactly matching the power of ripples during waking state and during nREM sleep. This result was already described in the Supplement (Supplementary Results 6.1). Second, we showed that replay levels during ripples are not correlated with ripple power, as one may expect if the power increase during ripples drives replay (Supplementary Results 6.2).

We now refer to these results more explicitly in the main manuscript on page 7 lines 18-21:

”After matching the ripple power between nREM sleep and waking state, the ripple-triggered replay effect was still restricted to nREM sleep (*Supplementary Results 6.1*). We also did not find any relation between gamma power and ripple-locked replay levels (*Supplementary Results 6.2*) which is consistent with our findings on spontaneous replay.”

These results show that replay levels do not directly relate to the magnitude of hippocampal gamma power. We obtained a similar result for our control analyses on spontaneous replay, showing that spontaneous replay was not related to the overall amount of gamma power averaged across all electrodes (Supplementary Result 4.2).

If replay were primarily driven by increases in power, it should be most pronounced at the peak frequency of the ripple-related power increase, i.e. at 80-100Hz. We tested this in an additional control analysis. First, we extracted the EEG power between 80-100Hz during each encoding trial and each ripple event. We then correlated the 80-100Hz power of each encoding time bin and each ripple event as in our main analysis of ripple-triggered replay. We performed a 2 x 2 ANOVA with “encoding cluster” and “subsequent memory” as repeated measures, again following the same procedure as reported in the manuscript (related to Figure 4D). The interaction effect of ripple-triggered replay during nREM sleep was no longer significant ($F(1,9)=0.94$, $p=0.36$; Fig. R4). We also did not find different replay levels during ripple events and peri-ripple periods for either remembered or forgotten items during either waking state or nREM sleep for the late encoding cluster (500-1,200ms; all $t(11)<1.60$, all $p>0.14$). These results further indicate that the high replay level during ripple events is not due to increased EEG power at the ripple frequency band.

Fig. R4 No interaction effect between “cluster” and “memory” of ripple-locked replay of the 80-100Hz EEG activities.

9). As a control, the authors could test replay in states of elevated gamma power that are not classified as ripples.

Response: Thanks for the suggestion. As described above, several control analyses show that replay levels are not directly driven by hippocampal power increases. Nevertheless, we followed the suggestion of the Referee and performed an additional control analysis in which we directly compared ripple-locked replay with replay during “surrogate ripple events”. Surrogate ripple events were extracted from channels other than the hippocampal channel that was used for detection of ripple events. For each iteration of the surrogate extraction, we randomly selected one channel for each participant and extracted surrogate ripple events following the same procedure as for the empirical ripple extraction. The EEG power (80-100Hz) of surrogate ripple events did not differ from the power of hippocampal ripple events ($t(11)=1.31$, $p=0.22$). In line with our main finding on spontaneous replay, we found that surrogate ripple-locked replay levels of both the early and late encoding activities were higher for remote than recent items both for remembered and forgotten items locked to surrogate ripples (all $t(9)>3.02$, all $p<0.012$). Importantly, however, when we compared replay during surrogate ripple events with surrogate peri-ripple periods during nREM sleep, we did not find any

difference for either the early or the late encoding activities, either for remembered or forgotten remote items (all $t(9)<1.16$, all $p>0.13$).

These results were added into the manuscript (Supplementary Results 5.2 and Fig. S8d).

Fig. R5 (Fig. S8d) Time resolved replay locked to surrogate ripples during nREM sleep and waking state for later remembered and forgotten remote items. Ripple time 0 corresponds to the time point of ripple events and ripple time before and after 0 corresponds to peri-ripple periods.

As a second control analysis, we also directly compared replays of remote items during empirical (hippocampal) ripple events with replay levels during surrogate ripple events. We first focused on replay of activity from the late encoding cluster during ripple events and compared it against zero by performing a one-sample T-test across participants. This generated one empirical $t_{\text{empirical}}$ value, which was then compared to the distribution of 10,000 surrogate $t_{\text{surrogate}}$ values. For remembered items during nREM sleep, we found that the $t_{\text{empirical}}$ value was higher than 95.23% (corresponding to $p=0.0477$) of the $t_{\text{surrogate}}$ values for the late encoding activities (500-1,200ms). For forgotten items, we observed a trend for replay of activity from the early encoding cluster ($p=0.0697$), similar to our result of enhanced replay during ripple events as compared to peri-ripple periods (p. 6). We performed the same analysis for replay of late encoding activities of forgotten items during nREM sleep and for replay of early and late encoding activities of remembered and forgotten items during waking state and did not find any significant results (all $p>0.1031$).

As a third control analysis, we selected epochs with elevated gamma power during rest satisfying two criteria: 1) the gamma power (30-90Hz) from electrodes in the hippocampus had to rank higher than 99% of the entire rest period; 2) the ripple power (80-100Hz) from the same electrode had to rank below the top 1% of the entire rest period (i.e., not fulfilling the criterion for a ripple event). By directly comparing replay levels of these selected epochs with replay levels within five epochs before and after the selected epochs, we found no difference for either the early or the late encoding activities during either nREM sleep or waking state for either remembered or forgotten items (all $t<1.41$, $p>0.20$). No interaction effect between clusters (early vs. late encoding activities) and memory performance (remembered vs. forgotten items) was found during either nREM sleep or waking state (both $F<0.50$, $p>0.62$). This result has been added to Supplementary Results 6.2.

10). Ripple specificity: The authors need to demonstrate that replay is not only enhanced in the peri-ripple time window (before and after), but also compare it to random epochs that are not being selected based on increased high gamma power.

Response: Thanks again for the suggestion. We performed the suggested analysis and compared ripple-locked replay in the empirical data with replay locked to randomly drawn surrogate events by matching the duration and incidents between ripple events and random epochs within each patient. First, ripple-locked nREM sleep replay of the encoding activities in the late cluster (Fig. 2a) was extracted for remembered remote items for each subject. We performed a one-sample T-test across subjects against zero (t_{ripple}). This was compared to t-values with the same contrast during randomly selected epochs. For each permutation, we selected random epochs by matching the duration and the number of ripple events of each subject. We calculated the replay levels within these randomly selected epochs and performed the same one-sample t-test against zero. The permutation procedure was repeated 10,000 times and generated 10,000 t_{random} values. We then ranked t_{ripple} within the distribution of t_{random} and found that ripple-locked replay was indeed significantly higher than during randomly selected epochs ($p=0.0453$).

We next applied the same analysis to forgotten remote items during nREM sleep, as well as to remembered and forgotten items during waking state, for both the early and the late encoding cluster. In line with our results in the main manuscript, we found that ripple-locked replay of the early cluster was higher than chance specifically for forgotten remote items during nREM sleep ($p=0.0157$). No other empirical ripple-locked replay was higher than chance (all $p>0.4282$).

These results are now described in detail in the Supplementary Results 5.1.

11). How do RSA values change with? Is there a general trend in the data that is independent of wake vs. sleep?

Response: We assume the Referee was asking whether RSA values change with time. We addressed this question when we assessed temporal proximity effects in our data and did not observe any significant linear trends (the data was reported in the Supplementary Results 4.3; Fig. S6d). In other words, we did not find more replay of remote items during early vs. late sleep periods or more “replay” of recent items during late vs. early sleep periods:

Fig. R6 (Fig. S6d) Replay is not related to temporal proximity between encoding and rest. Colored circles indicate results of individual subjects. Circles with the same colors indicate data from the same subject.

12). Did the authors consider time-compressed replay?

Response: We thank the Referee for this interesting suggestion. We checked both temporally-compressed and temporally-expanded replay by using the following scale parameters (scale parameter = encoding time length/rest time length): 0.2, 0.4, 0.6, 0.8, 1.2, 1.4, 1.6, 1.8, and 2.0. We found that the replay level of remote items was higher than the replay level of recent items with all scale parameters for the late encoding cluster (all $t(11) > 2.31$, all $p < 0.044$). We also did not find any difference between temporally-compressed/expanded replay and replay levels at original temporal scale (all $t(11) < 0.89$, $p > 0.38$). Since our data show substantial temporal autocorrelations within intervals of up to 250ms, we believe that these results need to be interpreted with caution and should be addressed in future studies.

We have added a corresponding sentence to the Results section on page 4, line 2 37-28: “We did not find any evidence for temporally-compressed or expanded replay effects (*Supplementary Results 3.5*).”

We describe these results in more detail in *Supplementary Results 3.5*:

“We also considered possible temporally compressed (or expanded) replay by using the following scale parameter (scale parameter = encoding time length/rest time length): 0.2, 0.4, 0.6, 0.8, 1.2, 1.4, 1.6, 1.8, and 2.0. We found that the replay level of remote items was higher than the replay level of recent items with all scale parameters for the late encoding cluster (all $t(11) > 2.31$, all $p < 0.044$). We did not find any difference between temporally-compressed/expanded replay to replay levels at the original temporal scale (all $t(11) < 0.89$, $p > 0.38$). Since our data show substantial temporal autocorrelation within intervals of up to 250ms (*Supplementary Results 2.3*; *Supplementary Figure S2d*), we believe that these results need to be interpreted with caution and should be addressed in future studies.”

Taken together, the authors present an impressive list of analyses. Some findings are interesting to broad range of neuroscientists (e.g. behavioral relevance during encoding in Fig. 2), but there is only little evidence for ripple-triggered replay. In addition, it seems that the described effects are not specific to NREM sleep, but also occur during wakefulness. A major caveat is that replay as studied here does not predict subsequent behavioral outcome. However, this paper adds to the existing literature and e.g. describes differential replay effects of remembered and forgotten items in terms of their relative timing to each other (Fig 4d).

Response: We are glad to read this overall positive assessment. We hope that our clarifications and additional results convince the Referee that there is also evidence for ripple-triggered replay. However, we would also be willing to tone down our conclusions on this part of our data if requested.

Reviewer #2 (Remarks to the Author):

In this study, Zhang et. al. use intracranial recordings from patients with epilepsy and identify stimulus elicited changes in gamma power which tend to replay during post encoding sleep and wakeful rest. They report high levels of spontaneous replay of neural activity, measured during the early and late encoding phases, in sleep and rest. Levels of spontaneous replay were not different when comparing the later remembered trials versus the forgotten ones in the post sleep retrieval task. The neural representations of remembered and forgotten items have higher levels of replay of their late and early encoding phases respectively, during ripple events compared to peri-ripple windows. This is only seen in nREM and not waking state, further supporting the idea that sleep is essential for memory consolidation.

Critique

This paper tackled questions that can elucidate mechanisms of memory replay and consolidation in humans. It provides electrophysiological evidence for the occurrence of spontaneous and ripple-triggered replay of stimulus-elicited changes in gamma power. The authors parsed out the difference in replay of the early vs. late time durations of stimulus locked responses and relate those differences to memory performance. They also differentiated between replay during sleep and wakeful rest, and distinguished between both spontaneous and ripple-triggered replay. They used appropriate controls and addressed some confounds. However, there are few important points that need to be addressed:

1) The paper lacks discussion on the spatial dynamics of replay of stimulus specific representation. The paper could have identified the functional regions, which contributed most to their findings both at the group and single subject level.

Response: Thanks for this suggestion. In order to address this issue, we conducted a jackknife analysis to evaluate the contribution of different brain regions to stimulus-specific representations during the early and late encoding time cluster. This was done separately for each electrode (i.e. at the single subject level) because of the different implantation schemes across patients.

First, we calculated the overall representational similarity indices of each encoding cluster by averaging Fisher-Z-transformed correlation values across all time bins within each cluster for each stimulus (RSA_{total}). Second, we calculated the correlation between encoding and retrieval by leaving one electrode out ($E[i]$). Again, we calculated the representational similarity indices of $E[i]$ by averaging Fisher-Z-transformed correlation values across all bins within each cluster for each stimulus ($RSA_{E[i]}$). We calculated the contribution of each electrode by performing a paired T-test between RSA_{total} and $RSA_{E[i]}$ across all stimuli. The higher the difference between RSA_{total} and $RSA_{E[i]}$, the higher was the contribution of $E[i]$.

First, we found that about 15% of all electrodes contributed positively to stimulus-specific representations of either the early or the late encoding cluster, which is in line with our previous findings (Zhang et al., 2015). We further investigated the spatial distribution of positively contributing electrodes. As indicated in the following figure, we found that more electrodes showed significant positive contributions to the early encoding cluster than the late encoding cluster ($\chi^2(1) = 5.69, p = .017$) in the lateral temporal lobe (including fusiform gyrus, inferior temporal gyrus, and middle temporal gyrus). We found a trend for an opposite pattern for electrodes within the medial temporal lobe (MTL, including the hippocampus and the parahippocampal gyrus) where more electrodes contributed significantly to the late cluster than to the early cluster ($\chi^2(1) = 2.96, p = 0.085$). Due to the overall small number of electrodes in the

occipital (2 electrodes) and parietal (2 electrodes) cortex, we did not perform statistical analyses in these regions.

The results indicate that during the early encoding period (which is not related with memory performance), fusiform gyrus and inferior and middle temporal gyrus are more involved in processing information. On the other hand, the MTL seems to be more involved during the late encoding period which is related to memory performance. Interestingly, regions showing a higher contribution to the early encoding cluster are substantially overlapping with the ventral visual stream (a pathway for visual object recognition) proposed by [18], while the MTL is a key region for high level information processing and memory [19, 20]. This result is generally consistent with previous findings and theoretical frameworks such as the Levels of Processing theory [21] and supports our interpretation that activity during the early encoding cluster plays a role in more superficial (perceptual) processing, while activity during the late encoding cluster reflects deeper processing.

We have added these results in the revised version of our manuscript. In the methods, we now write (page 12, line 39 to page 13, line 12):

“Identification of brain regions supporting early and late stimulus-specific representations.

We conducted a jackknife analysis to evaluate the contribution of different brain regions to stimulus-specific representations within the temporal clusters obtained from the analysis above. This was done separately for each electrode (i.e. at the single subject level) because of the different implantation schemes across patients.

First, we calculated the overall representational similarity indices of each cluster by averaging Fisher-Z transformed correlation values across all time bins within each cluster for each stimulus (RSA_{total}). Second, we calculated the correlation between encoding and retrieval by leaving one electrode out (E/i). Again, we calculated the representational similarity indices of E/i by averaging Fisher-Z transformed correlation values across all bins within each cluster for each stimulus ($RSA_{E/i}$). We calculated the contribution of each electrode by performing a paired T-test between RSA_{total} and $RSA_{E/i}$ across all stimuli. The higher the difference between RSA_{total} and $RSA_{E/i}$, the higher was the contribution of E/i .”

In the results section, we added (page 3, lines 28-39; see also Fig. S2c):

“We further investigated the brain regions underlying stimulus-specific representations in both time clusters. We found that 15% of all electrodes contributed significantly to stimulus-specific representations of either the early or the late encoding cluster (Fig. S2c) which replicated our previous findings [3] supporting the theoretical notion of sparse coding. We further investigated the distribution of positively contributing electrodes and found that more electrodes showed significant positive contributions to the early encoding cluster than the late encoding cluster ($\chi^2(1) = 5.69, p = .017$) in the lateral temporal lobe (including fusiform gyrus, inferior temporal gyrus, and middle temporal gyrus). We found a trend for an opposite pattern for electrodes within the medial temporal lobe (MTL, including the hippocampus and the parahippocampal gyrus) where more electrodes contributed significantly to the late cluster than to the early cluster ($\chi^2(1) = 2.96, p = 0.085$). Due to the overall small number of electrodes in the occipital (2 electrodes) and parietal (2 electrodes) cortex, we did not perform statistical analyses in these regions.”

We also added in the discussion on page 8, lines 26-32:

“This assumption is supported by findings showing that electrodes located in fusiform and inferior temporal lobe are more involved in stimulus-specific representations of early encoding activities, while the medial temporal lobe (MTL) seems to be more engaged during representations of late encoding activity. Regions contributing to the early encoding cluster were substantially overlapping with the ventral visual stream [18] that is predominantly relevant for processing perceptual information, while the MTL is a key region for high level information processing and memory [20].”

Fig. R7 (Fig. S2c). *Left: Distribution of electrodes showing significant positive contributions to the early and the late encoding clusters showing stimulus-specific representations. Right: The percentage of electrodes showing positive contribution to the early and the late encoding clusters in MTL and the lateral temporal lobe. FL: frontal lobe; PL: parietal lobe; OL: occipital lobe; LTL: lateral temporal lobe; MTL: medial temporal lobe; * indicates $p < 0.05$; x indicates $p = 0.085$.*

2) The paper doesn't address a potential confound regarding the number of ripples detected in or the duration of recording of wakeful state which may explain why their results don't show that awake replays are related to memory performance.

Response: Thanks for the comment. In fact, we did not find different numbers of ripple events during nREM sleep as compared to waking state ($t(11)=1.37$; $p=0.20$). Similarly, we did not find that the duration of nREM sleep and waking state differed ($t(11)=0.82$; $p=0.43$) – again not surprising because the patients only conducted an afternoon nap, and periods of awake resting state before and after their sleep (and sometimes brief awakenings between sleep periods) were included in the analysis.

In the revised version of the manuscript, we describe this explicitly on page 4, lines 8-9 and on page 5, lines 35-36.

Page 4, lines 8-9: “we focused on waking state and nREM sleep (including sleep stages N1, N2 and N3). The duration did not differ between waking state and nREM sleep ($t(11)=0.82$; $p=0.43$).”

Page 5, lines 36-37: “The number of ripple events did not differ between waking state and nREM sleep ($t(11)=1.37$; $p=0.20$).”

3) This study is not (as claimed) the first electrophysiological evidence for replay or ripple-triggered replay in humans (see Jiang et al., Replay of large-scale spatio-temporal patterns from waking during subsequent NREM sleep in human cortex. Sci Reports 2017).

Response: Thanks for pointing us to this paper, which we agree is relevant in the context of our study. However, we would like to point out that while both studies investigated “replay”, this was related to very different events: While the study by Jiang and

colleagues identified “Motifs” (i.e., sequences of high-gamma activity distributed across electrodes) from longer resting periods containing various everyday activities and cognitive experiments, we extracted stimulus-specific representations during specific processing stages of individual stimuli while the patients conducted a typical recognition memory paradigm. Both approaches have complementary advantages and disadvantages – while the study by Jiang et al. may be considered more ecologically valid, our study allowed us to test more specifically replay of stimulus-specific activity from experimentally defined individual items (and even across earlier vs. later processing stages of these items). Moreover, we could relate replay levels to later recognition and therefore to memory consolidation (see also the statement in the Abstract of the Jiang study “[...] our studies provide no direct demonstration that the replay of Motifs contributes to consolidation.”).

We have now described the similarities and differences between the two studies in the Discussion section (page 9):

“A previous intracranial EEG study investigated replay of distributed sequences of high-gamma activity during sleep [17]. This study reported that replay could indeed be detected (i.e., it was enhanced during sleep periods following as compared to preceding the initial occurrence of these sequences) and that it was related to the timing of several sleep graphoelements including sleep spindles and hippocampal ripples. However, this previous study extracted activity sequences across waking periods characterized by various everyday activity and cognitive experiments, but not locked to specific events. By contrast, we identified stimulus-specific neural representations during a recognition memory task. This allowed us not only to track the re-occurrence of neural activity patterns across multiple processing stages of individual events, but also to relate replay to later memory of individual items and thereby to clarify the functional relevance of replay for memory consolidation.”

4) This paper could potentially contribute to linking extensive electrophysiological reactivation studies in rodents and fmri studies in humans. However, the link is not straightforward due to some critical differences in experimental design, such as the absence of pre-behavior sleep baseline in the present study.

Response: We agree that it would have been ideal to include a pre-experimental sleep period. However, this may have introduced a bias between the first and the second sleep period because of the typical physiological adaptations to sleep laboratory conditions. In addition, we did not want to overstrain the patients by fixing polysomnography electrodes two times. On the other hand, two sleep periods back in back with a learning session in between was logistically not possible because we used an experimental design with an afternoon nap, and patients would not have been able to sleep both before and after the experiment. We chose this approach because afternoon naps are known to benefit memory consolidation and because the presentation of stimuli both before and after the sleep period optimally balances temporal distances and thereby prevents any possible problems related to temporal autocorrelations. For the same reason, we had used a similar design in a previous EEG/fMRI study [1]. Nevertheless, we think that it would be interesting to compare our results to findings obtained in an experiment in which a pre-experimental control night is used.

This is now discussed on page 7: “Notably, in the rodent literature, researchers typically record activities from a control sleep period before the task [22]. However, in humans a bias between the first and the second sleep period may occur because of the typical

physiological adaptations to sleep laboratory conditions. In addition, we did not want to overstrain the patients by fixing polysomnography electrodes two times. On the other hand, two sleep periods back in back with a learning session in between was logistically not possible because we used an experimental design with an afternoon nap, and patients would not have been able to sleep both before and after the experiment. Instead of a control nap before the experiment to which replay during the experimental nap would be compared, we presented stimuli both before and after the nap and compared replay of remote and recent items. The presentation of stimuli both before and after the sleep period optimally balances temporal distances and thereby prevents any possible problems related to temporal autocorrelations. Further studies are required to assess the difference between those two experimental settings.”

5) There were many analyses that were not cohesively described in the methods section, and several figures that did not have sufficient caption information. Please see suggestions below on how to address these points.

Response: We apologize for the lack of clarity and thank you for these suggestions. We have changed the manuscript accordingly.

Major comments

6). The method described on Page 12 (first paragraph) is similar to template matching method for detecting replay in rodent literature (e.g. Louie and Wilson, 2001), but the results would be more convincing if the replay levels for real gamma temporal pattern were compared to the distribution of replay levels for surrogate patterns (e.g. constructed by shuffling the pattern time bins).

Response: Thanks for this suggestion. In principle, we agree that our approach is similar to a template matching method as described in the rodent literature. For example, in the study by [23], a template was used that consisted of the sequence of place cells firing at different place fields. In the present study, a comparable template would be the temporal pattern of gamma activity of successive stimuli. As suggested by the reviewer, we calculated the replay of the gamma pattern during sequences consisting of five successive stimuli (i.e., stimulus 1-5, stimulus 2-6 etc.). First, we extracted the trajectory of gamma activity during the late encoding cluster for each stimulus, and then concatenated sequences of five successive stimuli into one “template”. We then calculated the replay of each template by correlating it with brain patterns during rest across time and channels. The Fisher-Z transformed correlation values were averaged across templates and rest periods. We performed a one-sample t-test of the replay levels against zero and this resulted in an empirical t-value which was compared with a distribution of surrogate values. Surrogate data were generated by shuffling the order of items within the template (keeping the sequence of activity from each item constant). During each permutation, we correlated the surrogate template with rest periods as for the empirical data, resulting in one surrogate t-value. The permutation procedure was repeated 1,000 times, resulting in 1,000 surrogate t-values. We then ranked our empirical t-value within the distribution of surrogate t-values. We found that the empirical t-value was larger than 84.7% of surrogate t-values, corresponding to $p=0.153$.

7). In addition, replay in rodent studies is typically time compressed and it would be interesting address this question in the present study.

Response: We thank the Referee for this suggestion. We checked both temporally-compressed and temporally-expanded replay by using the following scale parameters (scale parameter = encoding time length/rest time length): 0.2, 0.4, 0.6, 0.8, 1.2, 1.4, 1.6, 1.8, and 2.0. We found that the replay level of remote items was higher than the replay level of recent items with all scale parameters for the late encoding cluster (all $t(11) > 2.31$, all $p < 0.044$). We did not find any difference between temporally-compressed/expanded replay and replay levels at the original temporal scale (all $t(11) < 0.89$, $p > 0.38$). Since our data show substantial temporal autocorrelations within intervals of up to 250ms, we believe that these results need to be interpreted with caution and should be addressed in future studies.

We have added a corresponding sentence to the Results section on page 4, lines 27-28: “We did not find any evidence for temporally-compressed or expanded replay effects (Supplementary Results 3.5).”

We describe these results in more detail in Supplementary Results 3.5:

“Temporally-compressed or expanded replay in the gamma band. We also considered possible temporally compressed (or expanded) replay by using the following scale parameter (scale parameter = encoding time length/rest time length): 0.2, 0.4, 0.6, 0.8, 1.2, 1.4, 1.6, 1.8, and 2.0. We found that the replay level of remote items was higher than the replay level of recent items with all scale parameters for the late encoding cluster (all $t(11) > 2.31$, all $p < 0.044$). We did not find any difference between temporally-compressed/expanded replay to replay levels at the original temporal scale (all $t(11) < 0.89$, $p > 0.38$). Since our data show substantial temporal autocorrelations within intervals of up to 250ms (Supplementary Results 2.3; Supplementary Figure S2d), we believe that these results need to be interpreted with caution and should be addressed in future studies.”

8). What was the spatial distribution of electrodes for each patient? Authors should try and remove electrodes from each region at a time, see if significance still holds, and assess which electrodes contribute most to their results. It would be informative to know if there is a spatial pattern.

9). [Page 7 third paragraph line 7] The authors discussed that the early encoding cluster may play a role in more superficial, perceptual processing, and the late encoding cluster may play a role in deeper, more semantic processing. To provide better evidence for this, the authors can consider whether lower levels sensory areas such as the occipital or occipito-temporal electrodes contribute more to the early encoding gamma trace while higher level cortical association areas such as VTC, frontal, and parietal contribute more significantly to the later encoding phase.

Response: Thanks for the interesting suggestion. Because comments #8 and #9 are similar, we answer them together here. We performed a jackknife procedure as suggested (the procedure and results are also described in our reply to comment #1):

First, we found that about 15% of all electrodes contributed positively to stimulus-specific representations of either the early or the late encoding cluster, which is in line with our previous findings [3]. We further investigated the spatial distribution of positively contributing electrodes. As indicated in the following figure, we found that more electrodes showed significant positive contributions to the early encoding cluster than the late encoding cluster ($\chi^2(1) = 5.69$, $p = .017$) in the lateral temporal lobe (including fusiform gyrus, inferior temporal gyrus, and middle temporal gyrus). We

found a trend for an opposite pattern for electrodes within the medial temporal lobe (MTL, including the hippocampus and the parahippocampal gyrus) where more electrodes contributed significantly to the late cluster than to the early cluster ($\chi^2(1) = 2.96, p = 0.085$). Due to the overall small number of electrodes in the occipital (2 electrodes) and parietal (2 electrodes) cortex, we did not perform statistical analyses in these regions.

The results indicate that during the early encoding period (which is not related with memory performance), fusiform gyrus and inferior and middle temporal gyrus are more involved in processing information. On the other hand, the MTL seems to be more involved during the late encoding period which is related to memory performance. Interestingly, regions showing a higher contribution to the early encoding cluster are substantially overlapping with the ventral visual stream (a pathway for visual object recognition) proposed by [18], while the MTL is a key region for high level information processing and memory [19, 20]. This result is generally consistent with previous findings and theoretical frameworks such as the Levels of Processing theory [21] and supports our interpretation that activity during the early encoding cluster plays a role in more superficial (perceptual) processing, while activity during the late encoding cluster reflects deeper processing.

We have added these results in the revised version of our manuscript. In the methods, we now write (page 12, line 39 to page 13, line 12):

“Identification of brain regions supporting early and late stimulus-specific representations.

We conducted a jackknife analysis to evaluate the contribution of different brain regions to stimulus-specific representations within the temporal clusters obtained from the analysis above. This was done separately for each electrode (i.e. at the single subject level) because of the different implantation schemes across patients.

First, we calculated the overall representational similarity indices of each cluster by averaging Fisher-Z transformed correlation values across all time bins within each cluster for each stimulus (RSA_{total}). Second, we calculated the correlation between encoding and retrieval by leaving one electrode out ($E[i]$). Again, we calculated the representational similarity indices of $E[i]$ by averaging Fisher-Z transformed correlation values across all bins within each cluster for each stimulus ($RSA_{E[i]}$). We calculated the contribution of each electrode by performing a paired T-test between RSA_{total} and $RSA_{E[i]}$ across all stimuli. The higher the difference between RSA_{total} and $RSA_{E[i]}$, the higher was the contribution of $E[i]$.”

In the results section, we added (page 3, lines 28-39; see also Fig. S2c):

“We further investigated the brain regions underlying stimulus-specific representations in both time clusters. We found that 15% of all electrodes contributed significantly to stimulus-specific representations of either the early or the late encoding cluster (Fig. S2c) which replicated our previous findings [3] supporting the theoretical notion of sparse coding. We further investigated the distribution of positively contributing electrodes and found that more electrodes showed significant positive contributions to the early encoding cluster than the late encoding cluster ($\chi^2(1) = 5.69, p = .017$) in the lateral temporal lobe (including fusiform gyrus, inferior temporal gyrus, and middle temporal gyrus). We found a trend for an opposite pattern for electrodes within the medial temporal lobe (MTL, including the hippocampus and the parahippocampal

gyrus) where more electrodes contributed significantly to the late cluster than to the early cluster ($\chi^2(1) = 2.96, p = 0.085$). Due to the overall small number of electrodes in the occipital (2 electrodes) and parietal (2 electrodes) cortex, we did not perform statistical analyses in these regions.”

We also added in the discussion on page 8, lines 26-32:

“This assumption is supported by findings showing that electrodes located in fusiform and inferior temporal lobe are more involved in stimulus-specific representations of early encoding activities, while the medial temporal lobe (MTL) seems to be more engaged during representations of late encoding activity. Regions contributing to the early encoding cluster were substantially overlapping with the ventral visual stream [18] that is predominantly relevant for processing perceptual information, while the MTL is a key region for high level information processing and memory [20].”

10). [3 Page 11 Supplementary figure] The authors’ finding that replay in awake state is not related to memory performance is contrary to the rodent literature and the authors did not address this issue in sufficient detail. The limited number of ripples occurred during wakeful rest can explain their lack of finding. The authors should mention if sleep and wakeful rest durations were matched. The hypnograms presented in supplementary materials show that patients spent a much longer duration in sleep than in awake state. It is probable that there was insufficient wakeful rest data. The brain is also more dynamic during awake state and much longer durations may be necessary to ensure that the brain spends enough time in the state in which ripples occur.

Response: We agree that this is an important point. We have carefully checked the duration of the different stages of alertness and the number of ripple events during waking state and nREM sleep. In three of four example histograms (Fig. S3b), by chance, it seems that there is more nREM sleep than waking state. However, there are other subjects, where this would look the other way round. Importantly, on average there was no difference between the duration of nREM sleep and the duration of waking state (Fig. S3a). In the sleep stage averages across hypnograms (Fig. S3a), the duration of awake resting state is indeed longer than the duration of each individual nREM state (including sleep stages N1, N2 and N3; Fig. S3a). However, because of the overall relatively short duration of the nap, we collapsed across nREM sleep stages N1-N3, and the duration of this combined period of nREM sleep does not differ from the duration of awake resting state ($t(11)=0.82; p=0.43$). Also, the number of ripples during the combined nREM sleep periods does not differ from the number of ripples during awake resting state ($t(11)=1.37; p=0.20$; see also response to comment #2 above).

This is now described in greater detail on page 4, lines 8-9 and on page 5, lines 35-36.

Page 4, lines 8-9: “we focused on waking state and nREM sleep (including sleep stages N1, N2 and N3). The duration did not differ between waking state and nREM sleep ($t(11)=0.82; p=0.43$).”

Page 5, lines 36-37: “The number of ripple events did not differ between waking state and nREM sleep ($t(11)=1.37; p=0.20$).”

Nevertheless, we agree that the brain physiology is more dynamic during waking state than during sleep, so that the contribution of awake resting state may be more difficult to detect. This is now acknowledged in the *Supplementary Results 5.3*:

“Even though we did not find a contribution of ripples during awake resting state to replay, this may be explained by considering that the brain is more dynamic during waking state and therefore replay may be more difficult to detect.”

11). Rodent electrophysiology replay experiments typically use pre-behavior sleep as a baseline for comparison with post-behavior sleep. However, in this study, the authors compared replay of pre-sleep task with sleep, and that of post-sleep task with sleep. The experimental design in this study is different from the traditional approach and the authors should discuss their reasons for this experimental design.

Response: We agree that it would have been ideal to include a pre-experimental sleep period. However, this may have introduced a bias between the first and the second sleep period because of the typical physiological adaptations to sleep laboratory conditions. In addition, we did not want to overstrain the patients by fixing polysomnography electrodes two times. On the other hand, two sleep periods back in back with a learning session in between was logistically not possible because we used an experimental design with an afternoon nap, and patients would not have been able to sleep both before and after the experiment. We chose this approach because afternoon naps are known to benefit memory consolidation and because the presentation of stimuli both before and after the sleep period optimally balances temporal distances and thereby prevents any possible problems related to temporal autocorrelations. For the same reason, we had used a similar design in a previous EEG/fMRI study [1]. Nevertheless, we think that it would be interesting to compare our results to findings obtained in an experiment in which a pre-experimental control night is used.

This is now discussed on page 7: **“Notably, in the rodent literature, researchers typically record activities from a control sleep period before the task [22]. However, in humans a bias between the first and the second sleep period may occur because of the typical physiological adaptations to sleep laboratory conditions. In addition, we did not want to overstrain the patients by fixing polysomnography electrodes two times. On the other hand, two sleep periods back in back with a learning session in between was logistically not possible because we used an experimental design with an afternoon nap, and patients would not have been able to sleep both before and after the experiment. Instead of a control nap before the experiment to which replay during the experimental nap would be compared, we presented stimuli both before and after the nap and compared replay of remote and recent items. The presentation of stimuli both before and after the sleep period optimally balances temporal distances and thereby prevents any possible problems related to temporal autocorrelations. Further studies are required to assess the difference between those two experimental settings.”**

12). Authors are citing studies showing higher BOLD signal in areas involved in recent learning as evidence for replay in humans (e.g. Peigneux et al., 2004), and present their results as an electrophysiological demonstration of the same phenomena. Although BOLD signal is positively correlated with gamma power (Logothetis et al., 2001), replay scores and gamma power in their results are slightly negatively correlated. This discrepancy requires discussion.

Response: We agree that this result appears counterintuitive at first sight. Indeed, replay scores were numerically higher when gamma power during rest was lower. Even though this effect was not significant (and is therefore difficult to interpret), it is reminiscent of a previous finding from our group that encoding-retrieval similarity during viewing of movies is negatively related to gamma power during retrieval in a

significant subset of electrodes ([3], Figure 4). This was particularly the case for “informative” electrodes, i.e. those electrodes which contributed to the global representations (based on a jackknife procedure; see [3], Figure S4B). Our interpretation for this effect is that high levels of gamma power in the positively contributing (“informative”) electrodes obscure item-specific spatial patterns and thereby reduce reinstatement of information from encoding.

We have added a discussion of this point (page 8):

“Previous studies combining electrophysiological recordings with fMRI in monkey visual cortex reported a positive relationship between gamma power and BOLD responses [24]. Thus, it may appear surprising that we found slightly negative correlations between replay scores and gamma power during rest. Even though this relationship was not significant (and is therefore difficult to interpret), it is reminiscent of a previous finding from our group that encoding-retrieval similarity during viewing of movies is negatively related to gamma power during retrieval in a significant subset of electrodes [3]. This was particularly the case for “informative” electrodes, i.e. those electrodes which contributed to global representations (based on a jackknife procedure; see [3]). Our interpretation for this effect is that high levels of gamma power in the positively contributing (“informative”) electrodes obscure item-specific spatial patterns and thereby reduce reinstatement of information from encoding. However, more studies are necessary to elucidate the relationship between the magnitude of neural activity measured at various different levels of brain organization (from spike rates to local field potentials and frequency-resolved iEEG data to BOLD responses) to the representation of stimulus-specific information.”

13). In the terms of novelty, this is not the first electrophysiological evidence of replay or ripple-triggered replay in humans (see Jiang et al., Replay of large-scale spatio-temporal patterns from waking during subsequent NREM sleep in human cortex. *Sci Reports* 2017), nor the first demonstration of replay using population activity reflected in LFP signal (see Wilber et al., Laminar organization of encoding and memory reactivation in the parietal cortex. *Neuron* 2017). These studies should be cited and the strong novelty claim (“Our study, for the first time in humans, distinguished spontaneous replay from ripple-triggered replay”) should be revised accordingly.

Response: We agree that both studies are relevant. We have cited them and added a comparative discussion on page 9 (see also response to comment #3 above):

“A previous intracranial EEG study investigated replay of distributed sequences of high-gamma activity during sleep [17]. This study reported that replay could indeed be detected (i.e., it was enhanced during sleep periods following as compared to preceding the initial occurrence of these sequences) and that it was related to the timing of several sleep graphoelements including sleep spindles and hippocampal ripples. However, this previous study extracted activity sequences across waking periods characterized by various everyday activity and cognitive experiments, but not locked to specific events. By contrast, we identified stimulus-specific neural representations during a recognition memory task. This allowed us not only to track the re-occurrence of neural activity patterns across multiple processing stages of individual events, but also to relate replay to later memory of individual items and thereby to clarify the functional relevance of replay for memory consolidation.

Another related study [25] provided evidence that local field potential responses recorded via tetrodes in rat parietal cortex are tuned to specific egocentric movements

(specifically, to specific combinations of angular and linear velocity), that these responses are reactivated during post-experience sleep periods and that reactivation is linked to hippocampal ripples. These results demonstrate that replay does not only occur at the level of single cells but also on a population level, conceptually similar to the results reported in our current study. This study differs from our study not only in terms of methodology (rats vs. humans, distributed iEEG representations vs. “modules” in parietal cortex, assessment of reactivation etc.), but also with regard to the fact that our design allowed us to assess the functional relevance of reactivation for subsequent memory. Nevertheless, we feel that the studies by Jiang and colleagues [17], Wilber et al. [25] (and the results presented here) constitute an important step towards an integration of animal and human research, which should be followed by future investigations in both rodents and humans with parallel experimental designs, recording methodology and analysis methods.”

As suggested, we have also changed the cited statement in our manuscript on page 9, lines 32-33. We now write: “Our study provides first evidence for a differential contribution of spontaneous and ripple-triggered replay to memory consolidation of stimulus-specific representations in humans.” We also modified sentences in Abstract and Introduction accordingly.

14). [Page 20 figure 4C] Figure 4C shows results specific for the remembered and forgotten trials. These trials consist of two types of stimuli, landscapes and buildings. It should be mentioned if the number of stimuli from each category is matched across forgotten and remembered trials as this may be a confound. For example, building trials may be more easily distinguishable by the patient, and therefore the higher performance may not be due to the intra-ripple replay of the later encoding cluster.

Response: Thanks for raising this point. It is indeed important to exclude that the memory effect of ripple-triggered replay is not due to enhanced replay of one of the two categories of items. We assessed whether there was a possible difference of ripple-locked replay between buildings and landscapes. We therefore compared for each encoding time bin ripple-locked relay levels between buildings and landscapes. We did not find any difference either during nREM sleep or waking state (all $p > 0.13$).

15) In addition, it is not mentioned if the images shown are matched for color, size, and intensity. There are many electrodes in the temporal lobe and ventral visual pathway, which respond selectively to certain features of visual stimuli.

Response: We compared (a) remote and recent images and (b) remembered and forgotten images with regard to color and intensity (the size of all images in the study is the same, namely 640 x 480 pixels). Color/Intensity were assessed by averaging the RGB values/intensity values across pixels (separately for R, G, and B) and then compared between the different conditions. We did not find that (a) remote and recent items or (b) remembered and forgotten items differed with regard to color or intensity (all $t < 1.79$, all $p > 0.10$).

This is now reported in the Supplementary Results 1:

“We compared (a) remote and recent images and (b) remembered and forgotten images with regard to color and intensity (the size of all images in the study is the same, namely 640 x 480 pixels). Color and intensity were assessed by averaging the RGB (separately for R, G, and B) and intensity values across pixels and then compared between the

different conditions. We did not find that (a) remote and recent items or (b) remembered and forgotten items differed with regard to color or intensity (all $t < 1.79$, all $p > 0.10$.)”

16). [Page 20 figure 4C] The authors compared remote (pre-sleep items) spontaneous replay to recent (post sleep items) replay, but not remote ripple triggered replay to that of recent. When comparing the remote spontaneous replay, it was not found to be statistically higher than that of recent in nREM sleep. Therefore, it should be confirmed that the ripple-triggered replay seen in nREM sleep of the remote items is significantly higher than baseline (recent items).

Response: Thanks for the suggestion. We directly compared ripple-triggered replay levels between remote and recent items for the late encoding cluster (showing functionally relevant stimulus-specific representations) during nREM sleep. First, we extracted Fisher-Z transformed correlation values during ripple events and then averaged across all trials during nREM sleep, separately for remote and recent items. Then we performed a paired T-test between remote and recent times across participants and generated an empirical t-value which was then compared to a distribution of surrogate t-values. The surrogate data was generated by randomly switching condition labels (i.e., recent and remote items) and then performing the same paired T-test for each iteration of the surrogate procedure. The procedure of generating the surrogate data was performed 10,000 times, resulting in 10,000 surrogate t-values. We then ranked the empirical t-value within the distribution of surrogate t-values.

We found that the ripple-triggered replay level of remote items was significantly higher than the ripple-triggered replay level of recent items during nREM sleep ($p = 0.016$). We applied the same analysis to ripple-triggered replay of the early cluster during nREM sleep and for both the early and late cluster during waking state. We found that the ripple-triggered replay level of remote items was significantly higher than for recent items also for the early cluster during nREM sleep ($p = 0.014$). We did not find any difference between remote and recent items during waking state for either cluster (both $p > 0.43$).

The results were added to the manuscript on page 6, lines 3-6: “In addition, the ripple-locked replay level of remote items was higher than the ripple-locked replay level of recent items for both the early and late cluster during nREM sleep (both $t(9) > 2.19$, both $p < 0.016$). This was not the case during waking state (both $t(11) < 0.81$, both $p > 0.43$.)”

17). The authors discussed the experimental design of nap vs. no nap day, and landscape vs. building stimuli. What is the purpose of using nap versus no nap days, and what is the reason for using two categories of visual stimuli? This should be discussed explicitly in the paper. It would also be interesting to know if patient accuracy was higher during the nap days.

Response: The study involving one nap day and one day without nap was originally designed to investigate the role of an afternoon nap on memory consolidation. Results from this original study in a smaller group of patients have been reported before [13, 26]. Performance did not differ between the nap day and the day without nap (accuracy: $t(11) = 0.70$; $p = 0.50$; response latency: $t(11) = 0.86$; $p = 0.41$). Originally, we used two different categories in order to investigate category-specific replay which, however, we felt is less mechanistic than item-specific replay. Moreover, stimuli from two different categories ensured attentive processing of the images by requesting subjects to indicate stimulus category by button presses.

**This is now explicitly reported in the manuscript on page 10, line 37 to page 11, line 3:
“A design with a nap day and a day without nap was chosen to investigate the role of an afternoon nap on memory consolidation. Results from this original study in a smaller group of patients than the current study have been reported in previous papers [13, 26]. We presented stimuli of landscapes and buildings. Stimuli from two different categories were used in order to ensure attentive processing of the images by requesting subjects to indicate stimulus category by button presses.”**

18). How was stimulus presentation duration determined? Did stimulus duration have a fixed interval, or was it based on the patient’s response time? If during encoding, the stimulus was still presented after the patient responded, then motor effects need to be accounted for in the neural signal. Additionally, response time should be discussed in relation to the time of elicited neural response. Lastly, please discuss whether patients were using their dominant hand for both button presses during encoding.

Response: Thanks for the comment. Each stimulus was presented for a fixed duration of 1,200 ms with a jittered inter-stimulus interval of 1800±200ms. The response latencies of remembered and forgotten items were not different (remembered mean±SD: 787±428ms; Forgotten mean±SD: 790±427ms; $t(11)=0.41$, $p=0.69$). Thus, differences between remembered and forgotten items cannot be explained by different response latencies.

We also addressed the question whether stimulus-specific encoding-retrieval similarity is related to motor effects. In our main analysis, we compared encoding-retrieval similarity of the same item with encoding-retrieval similarity of different items. Thus, if response times would be more similar between encoding and retrieval of the same item as compared to encoding of one and retrieval of a different item, this may affect our results. In order to address this possible confound, we calculated the correlation of the response latency between encoding and retrieval across the same items within each patient and then performed a one-sample T-test against 0 on the Fisher-Z transformed correlation value. This resulted in an empirical t-value which was compared to surrogate t-values. Surrogate data were generated by shuffling encoding trial labels within each patient and again performing the same one-sample T-test against 0. The permutation procedure was repeated 10,000 times and generated 10,000 surrogate t-values. We then ranked the empirical t-value within the distribution of surrogate t-values. We found that the empirical t-value was higher than 57.55% of surrogate t-values, corresponding to $p=0.42$. Thus, we did not find any consistent correlation of the response latency to same items between encoding and retrieval. Since stimulus-specific representations were analyzed using this contrast, our results cannot be explained by correlated motor responses.

All patients were right handed and responded with their dominant hand. We also calculated the distribution of electrodes showing positive contributions to stimulus-specific representations and found that 13.43% of electrodes from either hemisphere contributed to the representation of the late cluster. 16.81% of electrodes from the left hemisphere and 14.16% of electrodes from the right hemisphere contributed to the representation of the early cluster. There was no difference between left and right hemisphere for either the early or the late cluster (both $\chi^2(1) < 0.38$, $p > .54$). This result provides further evidence that the reactivation findings are not driven by the motor responses which would be expected to be related to activity in the left motor cortex.

19). It would be informative to clarify the size of peri-epileptic discharge window excluded from analysis and what proportion of trials/nap epochs were on average lost to exclusion?

Response: During encoding and retrieval, we excluded an entire trial if epileptic discharges occurred in this trial, resulting in exclusion of 14.22%±13.05% (mean±SD) of all trials. During the rest period, we excluded data in a peri-epileptic discharge window of 2 seconds before and after epileptic discharges, leading to exclusion of 6.86%±4.22% (mean±SD) of all data.

This is now described on page 11, lines 21-23 and page 13, lines 20-22 of the revised manuscript.

Page 11, lines 21-23 “EEG trials were visually inspected for artifacts (e.g., epileptiform spikes), and trials with artifacts were excluded from further analyses, resulting in exclusion of 14.22%±13.05% (mean±SD) of all trials.”

Page 13, lines 20-22: “[...] periods from 2 seconds before to 2 seconds after each artifact window were excluded from further analyses. This resulted in exclusion of 6.86%±4.22% (mean±SD) of all data.”

20). Minor comments

1. The main text and supplementary figures were not documented well enough by captions.
2. There is a typing error in the second paragraph of results in the third line ‘a with regard to encoding...’
3. There are several accidental font changes: third to last line on page 3 and last sentence of first paragraph on page 4, and several more
4. Missing space in page 2 line 3
5. Page 4, Line 11 and Page 6, Line 2. P values ~ 0.1 or ~0.07 should not be interpreted as indicating significant result, especially for a claim as strong as the higher replay levels of early gamma component predict lower memory performance. In addition, different p values are reported for what seems to be the same comparison (p = 0.071 on Page 6, Line 2 and p = 0.061 in Fig.4 caption, Page 20, Line 8).
6. Page 10, Line 16 and elsewhere. Matrixes should be matrices.
7. Page20, Fig.4A. Please note if the plot is centered at ripple onset (as it appears) or peak?
8. Page S12, Fig.S4 caption, Line 5. It seems like ‘x’ significance label is in the figure caption, but not on any of the plots.
9. Page S12, Fig.S4. For line plots, it would be helpful to use different colors for different lines, as well as shaded error bars.
10. Page S16, Fig.S8A. Time windows with significant difference should be denoted by symbol at the top of the plot, rather than using error bars of different color.

Response: Thanks for pointing out these issues. We have changed the manuscript accordingly.

Reviewer #3 (Remarks to the Author):

This manuscript, 'Electrophysiological mechanisms of human memory consolidation,' explores the interesting question as to whether hippocampal ripples as captured by intracranial EEG in patients with medically refractory seizures are linked to replay of neural activity during memory consolidation. This is a highly relevant study, and addresses an important question in the field. Previous evidence has suggested that activity present when items are encoded into memory are replayed. This happens both during retrieval, but also during times of memory consolidation. The authors begin their study here by demonstrating that investigating these replay events during a rest period following study (the rest period can either be when subjects are asleep or awake). The novelty of the manuscript is in demonstrating that these replay events are coincident with hippocampal replay during the rest period. This would be an important contribution, as it links previous rodent work demonstrating the presence of these ripples, and linking that work to human memory consolidation.

This is a good study, and the authors should be commended for both their analyses and their efforts. This would be a valuable addition to the literature, providing direct evidence of the role of hippocampal ripples in replay events. The manuscript and the analysis is a bit complicated, given the many comparisons between encoding time periods, remote and recent items, and awake and asleep states. Nonetheless, the results appear to be well supported.

Response: We thank the referee for this positive take on our manuscript.

I feel there are a few concerns that should be addressed.

Major concerns

1). The primary result rests on two major components. First, when examining the entire rest period, there appear to be greater replay events for remote stimuli compared to recent stimuli (a baseline control) and that replay activity is greater during ripple compared to peri-ripple time periods. Clearly, over the entire hour long rest period, there should only be a few time points during which replay events occur, and so the fact that there are greater levels of replay averaged over this entire period are remarkable given the amount of noise and replay independent activity that would occur during this time. I thought the comparison between ripple and peri-ripple time periods was interesting, but a more direct test to ask whether ripples mediate replay would be to examine whether the replay levels observed over the entire rest period actually return to baseline levels once those ripple time points are removed. In other words, rather than compare ripple time points to only the peri-ripple time periods, the authors should also demonstrate that removing the ripple time points results in a significant reduction in overall replay during rest. Or conversely, compare the replay during ripple events to replay during the remainder of the rest period.

Response: Thanks for this suggestion. We performed two control analyses as suggested by the referee.

Because ripples are relatively rare events as compared to the overall duration of the resting state (duration of ripple events vs. overall duration of the resting state = $1:292 \pm 74$; mean \pm STD), we performed the first control analysis by matching the duration and incidents between ripple events and rest epochs within each patient. First, we computed ripple-locked replay levels of the late encoding cluster (showing functionally relevant stimulus-specific representations) during nREM sleep for remembered remote items for

each participant and performed a one-sample t-test against zero across participants. This resulted in a t-value (t_{ripple}). We then compared t_{ripple} to a surrogate distribution of t-values using the same contrast but applied to randomly selected epochs. These epochs were selected by matching the duration and number of ripple events within each participant. We then calculated the replay levels within these randomly selected epochs for the late encoding cluster of remembered remote items within each patient and, again, performed a one-sample t-test against zero across participants. This again resulted in a t-value ($t_{\text{spontaneous}}$). We repeated the procedure of randomly selecting matched epochs for 10,000 times, resulting in 10,000 values for $t_{\text{spontaneous}}$. We then ranked t_{ripple} within the distribution of $t_{\text{spontaneous}}$ values. We found that t_{ripple} was higher than 95.47% of values of $t_{\text{spontaneous}}$ (corresponding to $p=0.0453$), consistent with our result showing that for remembered remote items the replay level of activity from the late encoding cluster was higher during ripple events than during peri-ripple periods during nREM sleep (on page 6).

We also performed the same analysis for ripple-triggered replay of activity from the early encoding time cluster for forgotten remote items during NREM sleep. We found a significantly higher replay as compared to the distribution of $t_{\text{spontaneous}}$ values ($p=0.0157$). This corroborates our result that for forgotten remote items the replay level of activity from the early encoding cluster was higher during ripple events than during peri-ripple periods during nREM sleep (on page 6).

By contrast, when we conducted the same analysis for replay of late encoding activity of forgotten items, for replay of early encoding activity of remembered items, or for replay during waking state, we did not find any significant effects (all t_{ripple} smaller than 58.06% of $t_{\text{spontaneous}}$). Ripple-locked replay of activity from the late encoding cluster of forgotten remote items during nREM sleep was even smaller than spontaneous replay levels (t_{ripple} smaller than 98.6% of $t_{\text{spontaneous}}$, corresponding to $p=0.014$).

In the second control analysis, we directly compared replay levels during ripple events with spontaneous replay level of the entire rest periods after segmenting the entire resting period into many small epochs lasting as long as the average duration of ripple events within each participant, excluding periods containing ripple events. Again we first analyzed replay of activity from the late encoding cluster for remembered items during nREM sleep. Replay levels during ripples and outside of ripples were averaged in each participant. We then performed a paired T-test across participants between ripple-locked replay and spontaneous replay. This generated an empirical t-value which we then compared to surrogate t-values. Due to the large difference of the number of data between ripple-locked replay and spontaneous replay, we estimated the significance level of the empirical t-value using surrogate t-values to rule out any effects caused by the largely different amount of data. Within each permutation, we collapsed ripple-locked replay and spontaneous replay within each participant and then randomly drew the same number of data as surrogate “ripple events”. We performed the same paired T-test across participants, resulting in a surrogate t-value. This surrogate procedure was repeated 10,000 times, resulting in 10,000 surrogate t-values. We then ranked the empirical t-value within the distribution of surrogate t-values. We found that the empirical t value was higher than 94.99% of surrogate t-values, corresponding to $p=0.0501$.

We applied the same analysis to activity from the late encoding cluster for forgotten items during nREM sleep, to both remembered and forgotten items during waking state,

and to activity from the early encoding cluster for both remembered and forgotten items during both waking and nREM sleep. We found that for forgotten items, ripple-locked relay level of activity from the early encoding cluster was higher than spontaneous replay levels during nREM sleep ($p=0.0037$). No other results were significant (all $p>0.11$).

The results of these two control analyses are highly consistent with each other and corroborate our main results showing enhanced ripple-triggered replay of late encoding activities for remembered items and of early encoding activities for forgotten items during nREM sleep.

These results were added to the manuscript on page 6, lines 18-21, page 6, lines 33-35 and Supplementary Results 5.1.

On page 6, lines 21-23 we write: “Moreover, the replay level of the late cluster was higher than both randomly selected epochs of spontaneous replay (Supplementary Result 5.1) and replay locked to surrogate ripples (Supplementary Result 5.2).”

On page 6, lines 33-35 we write: “The replay level of the early cluster was higher than both randomly selected epochs of spontaneous replay (Supplementary Result 5.1) and replay locked to surrogate ripples (Supplementary Result 5.2).”

Supplementary Results 5.1:

“Ripple-triggered replay vs. spontaneous replay. To investigate the enhanced ripple-locked replay during nREM sleep, we performed a comparison between the ripple-locked replay and the spontaneous replay. Because ripples are relatively rare events as compared to the overall duration of the resting state (duration of ripple events vs. overall duration of the resting state = $1:292 \pm 74$; mean \pm SD), we matched the duration and incidents between ripple events and rest epochs within each patient. First, we computed ripple-locked replay levels of the late encoding cluster (showing functionally relevant stimulus-specific representations) during nREM sleep for remembered remote items for each participant and performed a one-sample t-test against zero across participants. This resulted in a t-value (t_{ripple}). We then compared t_{ripple} to a surrogate distribution of t-values using the same contrast but applied to randomly selected epochs. These epochs were selected by matching the duration and number of ripple events within each participant. We then calculated the replay levels within these randomly selected epochs for the late encoding cluster of remembered remote items within each patient and, again, performed a one-sample t-test against zero across participants. This again resulted in a t-value ($t_{\text{spontaneous}}$). We repeated the procedure of randomly selecting matched epochs for 10,000 times, resulting in 10,000 $t_{\text{spontaneous}}$ values. We then ranked t_{ripple} within the distribution of $t_{\text{spontaneous}}$ values. We found that t_{ripple} was higher than 95.47% of values of $t_{\text{spontaneous}}$ (corresponding to $p=0.0453$), consistent with the result showing that the replay level of late encoding activities of remembered remote items during ripple events was higher than replay levels during peri-ripple periods during nREM sleep.

We also performed the same analysis for ripple-triggered replay of activity from the early encoding time cluster for forgotten remote items during NREM sleep and found a significantly higher replay as compared to the distribution of $t_{\text{spontaneous}}$ values ($p=0.0157$), again corroborating our results showing that the replay level of early encoding activities of forgotten remote items during ripple events was higher than replay levels during peri-ripple periods during nREM sleep.

By contrast, when we conducted the same analysis for replay of late encoding activity of forgotten items, for replay of early encoding activity of remembered items, or for replay during waking state, we did not find any significant effects (all t_{ripple} smaller than 58.06% of $t_{\text{spontaneous}}$). Ripple-locked replay of activity from the late encoding cluster of forgotten remote items during nREM sleep was even smaller than spontaneous replay levels (t_{ripple} smaller than 98.6% of $t_{\text{spontaneous}}$, corresponding to $p=0.014$).

We also directly compared replay levels during ripple events with spontaneous replay level of the entire rest periods after segmenting the entire resting period into many small epochs lasting as long as the average duration of ripple events within each participant, excluding periods containing ripple events. Again, we first analyzed replay of activity from the late encoding cluster for remembered items during nREM sleep. Replay levels during ripples and outside of ripples were averaged in each participant. We then performed a paired T-test across participants between ripple-locked replay and spontaneous replay. This generated an empirical t-value which we then compared to surrogate t-values. Due to the large difference of the number of data between ripple-locked replay and spontaneous replay, we estimated the significance level of the empirical t-value using surrogate t-values to rule out any effects caused by the largely different amount of data. Within each permutation, we collapsed ripple-locked replay and spontaneous replay within each participant and then randomly drew the same number of data as surrogate “ripple events”. We performed the same paired T-test across participants, resulting in a surrogate t-value. This surrogate procedure was repeated 10,000 times, resulting in 10,000 surrogate t-values. We then ranked the empirical t-value within the distribution of surrogate t-values. We found that the empirical t value was higher than 94.99% of surrogate t-values, corresponding to $p=0.0501$. We applied the same analysis to the late encoding activities for forgotten items during nREM sleep and both remembered and forgotten items during waking state and the early encoding activities for both remembered and forgotten items during both waking and nREM sleep. We found that the ripple-locked relay level of the early encoding activities was higher than the spontaneous replay level for forgotten items during nREM sleep ($p=0.0037$). No other results were significant (all $p>0.11$).”

2). The second major concern I have relates to the memory relevance of these ripple events. Most of the analyses focus on the late encoding period, and suggest that memory relevance is only present when examining activity from 500-1200 ms during encoding, even though the early time periods also demonstrated stimulus specific replay. The authors explain this result by suggesting that there is a deep encoding period that occurs that is relevant for memory formation that is distinct from superficial stimulus processing. However, as it relates to ripples, it appears that the only time periods that show a significant memory related modulation occur even later during encoding, as demonstrated in Figure 4e (around 1200 ms). It is difficult to understand, then, how the ripple related replay events that correspond to earlier time points are relevant for memory, and why such distinctions occur only at the very end of the encoding period.

Response: We agree that this is a relevant point. Indeed, a direct comparison of ripple-triggered replay levels of remembered and forgotten items revealed significant differences during a relatively late encoding time period, i.e. between 1,100-1,200ms.

Because of the 200ms width of the intervals we extracted, this period actually captures activity in the time range from 1,000 to 1,300ms.

The relatively late occurrence of this time window corresponds to a part of the late encoding time window which shows stimulus-specific representations that are related to subsequent memory (Figure 2A). This time period also overlaps with the temporal cluster showing a significant difference between spontaneous replay of remote and recent items (Figure 3e.i, cluster 3).

In order to investigate in greater detail which encoding time windows are relevant for later memory, we conducted an additional analysis in which we tested how memory effects (i.e., differences in encoding-retrieval similarity between later remembered and forgotten items) emerge across time. Indeed, we found that predominantly reinstatement of activity during late time windows (800 to 1,100ms; capturing activity in the time range from 700 to 1,200ms) is relevant for subsequent memory ($p < 0.0001$ after multiple comparison correction; surrogate data were generated by shuffling condition labels).

These additional results are now reported in the revised version of the manuscript on page. 3, lines 18-24: “In order to investigate in greater detail which encoding time windows are relevant for later memory, we compared the difference of the similarity levels between remembered and forgotten items within clusters showing stimulus-specific representations and found that predominantly reinstatement of activity during late time window (800 to 1,100ms; capturing activity in the time range from 700 to 1,200ms) was relevant for subsequent memory ($p < .001$ after multiple comparison correction; surrogate data were generated by shuffling condition labels of trial averages in individual participants).”

3). The authors present an analysis examining whether there is replay of the entire sequence. To do this, they construct larger concatenated matrices corresponding to 8 sequential time points, and examine the correlations of those larger matrices. However, to me, this appears to simply temporally smear the replay data that were analyzed using more temporally precise 200 ms window. In order to examine the sequence of activity, then an analysis should be performed on the trajectory of activity over this time window, rather than just the average correlation over the concatenated window. This is a supplementary point to the main results, in my view, and so unless a specific demonstration of how the activity progresses from time point to time point can be shown, it is hard to understand how this provides evidence of the sequence.

Response: Thanks for the comment. We calculated sequence replay as suggested here. We extracted the trajectory of gamma power within the encoding cluster showing stimulus-specific representation (500-1,200ms). To keep the temporal precision as suggested, we did not perform any further averaging of the gamma power. We then correlated these time courses with time courses of similar durations during the resting period. We found that the replay levels of remote items were higher than the replay levels of recent items during the concatenated rest period ($t(11)=2.35$; $p=0.038$). Replay levels of remote items were higher than those of recent items during waking state ($t(11) = 2.38$, $p=0.037$) but there was only a trend for an increase during nREM sleep ($t(9) = 2.17$, $p = 0.058$). The replay levels of remote items were significantly higher than zero, while this was not the case for recent items (again, we considered both replay during the entire rest period and separately for each vigilance state; remote: all $t(11) > 2.75$, all $p < 0.019$; recent: all $t(11) < 0.91$, all $p > 0.38$).

These results are now reported in Figure S5 and described in the Supplementary Results 3.2 and 3.4.

4). There was no evidence that there was a greater level of replay for remote compared to recent items during nREM sleep. Interestingly, the only differences occurred when comparing remote to recent replay during wake. Yet ripple events in their analysis were focused on nREM sleep. This point requires some clarification. Do ripple events only occur during sleep in their analysis (they do not show the distribution of ripple events during the different states of vigilance)? If so, then how should we interpret the fact that nREM sleep does not contain more replay events for remote compared to recent items.

Response: Thanks for raising this important point. Actually, ripple events occurred during both nREM sleep and waking state, and the number of events did not differ between these two states ($t(11)=1.37$; $p=0.20$). Thus, the fact that ripple-triggered replay occurs only during NREM sleep but not during awake resting state cannot be explained by the smaller number of ripple events during waking state.

This is now described more explicitly in the text on page 5, lines 36-37: “The number of ripple events did not differ between waking state and nREM sleep ($t(11)=1.37$; $p=0.20$).”

Thus, while spontaneous replay occurs more prominently during waking state, ripple-triggered replay is specific to nREM sleep. Even though these results may appear surprising at first sight, they actually provide an explanation of how two different lines of research may be integrated: On the one hand, several fMRI studies have described that spontaneous replay of stimulus-specific encoding activity can be observed during awake resting state [1, 2], and that awake replay is even more pronounced than replay during sleep [1]. On the other hand, abundant studies provide evidence for the beneficial role of sleep for memory consolidation [5, 6]. Thus, some sleep-specific effects other than spontaneous replay seem to contribute to consolidation, and our data suggest ripple-triggered replay as a strong candidate for this sleep-specific effects.

5). For the cluster analysis to correct for multiple comparisons, the authors state that they compare the empiric cluster to all clusters derived from each permutation. There are two points that should be clarified. First, the clusters used to derive the null distribution should only be comprised of the maximum cluster statistic from every permutation, not all possible clusters (as an individual permutation may have more than one significant cluster).

Response: We apologize for the lack of clarity. This is exactly what we did and have described on page 12, lines 22-23:” We extracted clusters from these surrogate matrices and selected the cluster with the largest summary t-value [...] for each surrogate”. However, this information has previously not been included in the figure legends. Thus, we now have added this information to the legend of Fig 1.

6). Second, it is not clear at what level the permutations are being conducted. If comparing across subjects, the permutations should be performed at the level of individual subjects, not the level of individual trials, so that the subject is the unit of observation. This would result in 2^{12} possible permutations (given 12 subjects) when comparing one condition to another, of which they could choose just 10,000 to make their null distribution. It is possible that this permutation was indeed performed at the level of subjects, but this is not clear.

Response: Thanks for raising this point. For assessing stimulus-specific representations during encoding, we compared the similarity between encoding and retrieval of the same item with the similarity between encoding of one and retrieval of a different item. These clusters were significant both compared to surrogates obtained from shuffling condition levels across individual trials, and compared to surrogates obtained from shuffling labels across trial averages (i.e. across subjects as suggested by the Referee).

This information has been added to the manuscript.

On page 3, lines 23-24 we write: “surrogate data were generated by shuffling condition labels of trial averages in individual participants”

On page 4, line 34 we write: “surrogate data was generated by switching remote and recent labels”

Please see also p. 5, lines 1-2: “surrogate data was generated by switching remote and recent labels”

And see page 12, lines 31-33: “We also utilized another method to extract surrogate clusters in which we randomly switched condition labels between RSA_{same} and RSA_{differ} at the subject level (i.e. across trial averages). This method generated the same results.”

References

1. Deuker, L., et al., *Memory consolidation by replay of stimulus-specific neural activity*. J Neurosci, 2013. **33**(49): p. 19373-83.
2. Staresina, B.P., et al., *Awake reactivation predicts memory in humans*. Proc Natl Acad Sci U S A, 2013. **110**(52): p. 21159-64.
3. Zhang, H., et al., *Gamma power reductions accompany stimulus-specific representations of dynamic events*. Curr Biol, 2015. **25**(5): p. 635-40.
4. Yaffe, R.B., et al., *Reinstatement of distributed cortical oscillations occurs with precise spatiotemporal dynamics during successful memory retrieval*. Proc Natl Acad Sci U S A, 2014. **111**(52): p. 18727-32.
5. Rasch, B. and J. Born, *About sleep's role in memory*. Physiol Rev, 2013. **93**(2): p. 681-766.
6. Schonauer, M., et al., *Exploring the effect of sleep and reduced interference on different forms of declarative memory*. Sleep, 2014. **37**(12): p. 1995-2007.
7. Diekelmann, S., et al., *Labile or stable: opposing consequences for memory when reactivated during waking and sleep*. Nature Neuroscience, 2011. **14**(3): p. 381-386.
8. Yaffe, R.B., et al., *Cued Memory Retrieval Exhibits Reinstatement of High Gamma Power on a Faster Timescale in the Left Temporal Lobe and Prefrontal Cortex*. J Neurosci, 2017. **37**(17): p. 4472-4480.
9. Cairney, S.A., et al., *Memory Consolidation Is Linked to Spindle-Mediated Information Processing during Sleep*. Current Biology, 2018. **28**(6): p. 948-+.
10. Jackson, J.C., A. Johnson, and A.D. Redish, *Hippocampal sharp waves and reactivation during awake states depend on repeated sequential experience*. J Neurosci, 2006. **26**(48): p. 12415-26.
11. O'Neill, J., T. Senior, and J. Csicsvari, *Place-selective firing of CA1 pyramidal cells during sharp wave/ripple network patterns in exploratory behavior*. Neuron, 2006. **49**(1): p. 143-55.
12. Staba, R.J., et al., *High-frequency oscillations recorded in human medial temporal lobe during sleep*. Ann Neurol, 2004. **56**(1): p. 108-15.
13. Axmacher, N., C.E. Elger, and J. Fell, *Ripples in the medial temporal lobe are relevant for human memory consolidation*. Brain, 2008. **131**(Pt 7): p. 1806-17.
14. Bagshaw, A.P., et al., *Effect of sleep stage on interictal high-frequency oscillations recorded from depth macroelectrodes in patients with focal epilepsy*. Epilepsia, 2009. **50**(4): p. 617-28.

15. Buzsaki, G. and X.J. Wang, *Mechanisms of gamma oscillations*. Annu Rev Neurosci, 2012. **35**: p. 203-25.
16. Staresina, B.P., et al., *Hierarchical nesting of slow oscillations, spindles and ripples in the human hippocampus during sleep*. Nat Neurosci, 2015. **18**(11): p. 1679-86.
17. Jiang, X., et al., *Replay of large-scale spatio-temporal patterns from waking during subsequent NREM sleep in human cortex*. Sci Rep, 2017. **7**(1): p. 17380.
18. Goodale, M.A. and A.D. Milner, *Separate visual pathways for perception and action*. Trends Neurosci, 1992. **15**(1): p. 20-5.
19. Eichenbaum, H. and P.A. Lipton, *Towards a functional organization of the medial temporal lobe memory system: role of the parahippocampal and medial entorhinal cortical areas*. Hippocampus, 2008. **18**(12): p. 1314-24.
20. Felleman, D.J. and D.C. Van Essen, *Distributed hierarchical processing in the primate cerebral cortex*. Cereb Cortex, 1991. **1**(1): p. 1-47.
21. Nyberg, L., *Levels of processing: a view from functional brain imaging*. Memory, 2002. **10**(5-6): p. 345-8.
22. Skaggs, W.E. and B.L. McNaughton, *Replay of neuronal firing sequences in rat hippocampus during sleep following spatial experience*. Science, 1996. **271**(5257): p. 1870-3.
23. Louie, K. and M.A. Wilson, *Temporally structured replay of awake hippocampal ensemble activity during rapid eye movement sleep*. Neuron, 2001. **29**(1): p. 145-56.
24. Logothetis, N.K., et al., *Neurophysiological investigation of the basis of the fMRI signal*. Nature, 2001. **412**(6843): p. 150-7.
25. Wilber, A.A., et al., *Laminar Organization of Encoding and Memory Reactivation in the Parietal Cortex*. Neuron, 2017. **95**(6): p. 1406-1419 e5.
26. Axmacher, N., et al., *The role of sleep in declarative memory consolidation--direct evidence by intracranial EEG*. Cereb Cortex, 2008. **18**(3): p. 500-7.

REVIEWERS' COMMENTS:

Reviewer #1 (Remarks to the Author):

The authors responded in great detail to all queries. They provided substantial new data, which further strengthened their conclusions and I'm happy to recommend publication.

Reviewer #2 (Remarks to the Author):

I'm in favor of this paper getting published. Their results contribute to our understanding of memory processing in humans, and they've connected their findings to both behavior and functional anatomy.

I'm particularly pleased with the following:

Their addressing of comments #1, 8 and 9 - identifying which functional regions contribute to their results. The regions they found to be contributive (lateral temporal lobe vs. MTL structures) are in line with their initial interpretations (perceptual versus memory processing) further supporting their claims.

They also addressed a second major concern about their lack of finding regarding awake state ripple contribution to memory [comment 10] - they did appropriate follow up analyses and no longer exclude this as a possibility.

They've addressed each comment extensively and investigated most suggested confounds.

I still have a few minor caveats, however, which the authors (at their discretion) may want to address:

- 1) They mention temporal autocorrelations as an argument against using the pre- vs. post-sleep contrast, while at the same time they go to lengths to downplay the impact of temporal autocorrelations on differences between the remote and recent item replay;
- 2) They show significant ripple-triggered replay for the range of compression/decompression factors (ranging from the sleep activity 5x slower to 2x faster than awake), which raises the question on what kind of activity underlies it.
- 3) Some of the key results show marginal statistical significance, which, for an analysis of this complexity, could be due to chance (e.g. $p \sim 0.045$).

Reviewer #3 (Remarks to the Author):

In this revised manuscript, the authors have clearly invested a significant amount of effort to address each of the points raised by the Reviewers. As far as my points are concerned, I believe that their responses are satisfactory and sufficient. Specifically, they have performed the requested control analyses to demonstrate that indeed ripple-locked replay is specific to the ripple events, and the analyses investigating which specific time windows are relevant for later memory. In addition, the analysis investigating the temporal sequence of replay using the concatenated data was nice. The authors do report that when performing their statistical analysis across participants, they observe similar results and significant differences. The only possible question I would have is that they actually report these results across subjects, but this is not a critical point.

Reviewer #1 (Remarks to the Author):

The authors responded in great detail to all queries. They provided substantial new data, which further strengthened their conclusions and I'm happy to recommend publication.

Response: We are glad to hear that referee #1 thinks that our manuscript is suitable for publication.

Reviewer #2 (Remarks to the Author):

I'm in favor of this paper getting published. Their results contribute to our understanding of memory processing in humans, and they've connected their findings to both behavior and functional anatomy.

I'm particularly pleased with the following:

Their addressing of comments #1, 8 and 9 - identifying which functional regions contribute to their results. The regions they found to be contributive (lateral temporal lobe vs. MTL structures) are in line with their initial interpretations (perceptual versus memory processing) further supporting their claims. They also addressed a second major concern about their lack of finding regarding awake state ripple contribution to memory [comment 10] - they did appropriate follow up analyses and no longer exclude this as a possibility.

They've addressed each comment extensively and investigated most suggested confounds.

I still have a few minor caveats, however, which the authors (at their discretion) may want address:

1) They mention temporal autocorrelations as an argument against using the pre- vs. post-sleep contrast, while at the same time they go to lengths to downplay the impact of temporal autocorrelations on differences between the remote and recent item replay;

Response: As pointed out by the Referee, we found that temporal autocorrelations decreased to an average level at a temporal distance of 250ms (Supplementary Notes 4 and Supplementary Fig. 2d), which was substantially shorter than the shortest distance between remote/recent items and the rest period (around 15 minutes). As a result, the fact that the replay level of remote items is higher than the replay level of recent items cannot be explained by temporal autocorrelation.

In general, however, adopting a design with a night before and a night after encoding may lead to problems with temporal distance from the test session – for example, when items are only encoded in the evening, then the preceding night has a much higher temporal distance from this encoding session than the following (experimental) night. This is one reason why we adopted the design with one nap preceded and followed by two encoding sessions, and why we wrote in the Discussion *'The presentation of stimuli both before and after the sleep period optimally balances temporal distances and thereby prevents any possible problems related to temporal autocorrelations'*.

2) They show significant ripple-triggered replay for the range of compression/decompression factors

(ranging from the sleep activity 5x slower to 2x faster than awake), which raises the question on what kind of activity underlies it.

Response: We agree that this is a relevant question. However, we would be cautious to interpret the kind of activity underlying compressed/decompressed replay, because our data show substantial temporal autocorrelation within intervals of up to 250ms. We believe that clarifying the activity underlying compressed/decompressed replay is an important issue for future studies using recordings methods with lower temporal autocorrelations (e.g. action potential recordings via microelectrodes).

3) Some of the key results show marginal statistical significance, which, for an analysis of this complexity, could be due to chance (e.g. $p \sim 0.045$).

Response: We used the conventional criterion of $p < 0.05$ for all tests and only considered p-values smaller than 0.05 as significant (e.g. $p = 0.051$ in Supplementary Fig. 4e was not considered significant). We also reported the exact p-value (e.g., $p = 0.031$) instead of a general range (e.g., $p < 0.05$) to disclose the exact significance level. However, we of course cannot totally exclude that some of the results are due to chance.

Reviewer #3 (Remarks to the Author):

In this revised manuscript, the authors have clearly invested a significant amount of effort to address each of the points raised by the Reviewers. As far as my points are concerned, I believe that their responses are satisfactory and sufficient. Specifically, they have performed the requested control analyses to demonstrate that indeed ripple-locked replay is specific to the ripple events, and the analyses investigating which specific time windows are relevant for later memory. In addition, the analysis investigating the temporal sequence of replay using the concatenated data was nice. The authors do report that when performing their statistical analysis across participants, they observe similar results and significant differences. The only possible question I would have is that they actually report these results across subjects, but this is not a critical point.

Response: We are glad to hear that this referee also appreciates our revised manuscript. As described, we performed the surrogate analysis when identifying the stimulus-specific representations in two different ways: First, we shuffled picture labels at the level of individual trials within each patient. Second, we shuffled the condition labels (RSA_{same} and RSA_{differ}) after averaging across trials in each participant, as suggested by the referee. The surrogate procedure was repeated 10,000 times for each of the two approaches. We found that the empirical clusters were larger than all surrogate clusters using both surrogate methods, corresponding to $p < 0.001$. This is now more explicitly reported in the Methods section (p. 13):

“We also utilized another method to extract surrogate clusters in which we randomly switched condition labels between RSA_{same} and RSA_{differ} at the subject level (i.e. across trial averages). This method generated the same results (i.e., $p < 0.001$).”